# THE INTRIGUING EFFECTS OF FOCAL LOSS ON THE CALIBRATION OF DEEP NEURAL NETWORKS

## ABSTRACT

Miscalibration – a mismatch between a model's confidence and its correctness – of Deep Neural Networks (DNNs) makes their predictions hard for downstream components to trust. Ideally, we want networks to be accurate, calibrated and confident. Temperature scaling, the most popular calibration approach, will calibrate a DNN without affecting its accuracy, but it will also make its correct predictions underconfident. In this paper, we show that replacing the widely used cross-entropy loss with focal loss allows us to learn models that are already very well calibrated. When combined with temperature scaling, focal loss, whilst preserving accuracy and yielding state-of-the-art calibrated models, also preserves the confidence of the model's correct predictions, which is extremely desirable for downstream tasks. We provide a thorough analysis of the factors causing miscalibration, and use the insights we glean from this to theoretically justify the empirically excellent performance of focal loss. We perform extensive experiments on a variety of computer vision (CIFAR-10/100) and NLP (SST, 20 Newsgroup) datasets, and with a wide variety of different network architectures, and show that our approach achieves state-of-the-art accuracy and calibration in almost all cases.

## 1 INTRODUCTION

Deep neural networks have dominated computer vision and machine learning in recent years, and this has led to their widespread deployment in real-world systems (Cao et al., 2018; Chen et al., 2018; Kamilaris and Prenafeta-Boldú, 2018; Ker et al., 2018; Wang et al., 2018). State-of-the-art networks achieve high levels of accuracy for many tasks. However, many current multi-class classification networks in particular are poorly calibrated, in the sense that the probability values that they associate with the class labels they predict for different test samples overestimate the likelihoods of those class labels being correct in the real world. This is a major problem, since if networks are routinely overconfident, then downstream components cannot trust their predictions. The underlying cause is hypothesised to be that these networks' high capacity leaves them vulnerable to overfitting on the negative log-likelihood (NLL) loss they conventionally use during training (Guo* et al., 2017).

Given the importance of this problem, numerous suggestions for how to address it have been proposed. Much work has been inspired by early approaches from the pre-deep learning era such as Platt scaling (Platt, 1999), histogram binning (Zadrozny and Elkan, 2001), isotonic regression (Zadrozny and Elkan, 2002), and Bayesian binning and averaging (Naeini et al., 2015; Naeini and Cooper, 2016). As deep learning has become more dominant, works have begun to directly target the calibration of deep networks. For example, Guo et al. (Guo* et al., 2017) have popularised a modern variant of Platt scaling known as *temperature scaling*, which works by dividing a network's logits by a scalar $T > 0$ (learnt on a validation subset) prior to performing softmax. Temperature scaling has the desirable property that it can improve the calibration of a network without in any way affecting its accuracy. Mozafari et al. (Mozafari et al., 2018) noted the downsides of using cross-entropy loss with temperature scaling, and proposed an alternative loss called Attended-NLL that helps temperature scaling achieve better calibration. More recently, Shrikumar and Kundaje (Shrikumar and Kundaje, 2019) have proposed an extension to temperature scaling that adds class-specific bias parameters to help eliminate systematic bias when performing domain adaptation. Separately, Hendrycks et al. (Hendrycks et al., 2019) have studied the effects of pre-training (vs. training from scratch) on model robustness and uncertainty: they make the interesting observation that because long periods of training can cause a network to become miscalibrated, tuning a pre-trained network, which facilitates faster convergence, can seemingly lead to a more calibrated model. Notably, since their approach

complements temperature scaling, the two techniques can also be used together to achieve even better calibration overall.

Whilst temperature scaling's simplicity and effectiveness have made it a popular and state-of-the-art network calibration technique, it does have downsides. For example, whilst it scales the logits to reduce the network's confidence in incorrect predictions, this also slightly reduces the network's confidence in predictions that were actually correct. By contrast, Kumar et al. (Kumar et al., 2018) eschew temperature scaling altogether in favour of minimising a differentiable proxy for calibration error at training time, called Maximum Mean Calibration Error (MMCE). However, they also use temperature scaling as a post-processing step to obtain better results than cross-entropy followed by temperature scaling (Guo* et al., 2017).

In this paper, we propose a technique for improving network calibration that works by replacing the cross-entropy loss conventionally used when training multi-class classification networks with the focal loss proposed by Lin et al. (Lin et al., 2017) for dense object detection. Since focal loss, as shown in §4, is dependent on a hyperparameter, $\gamma$, which needs to be cross-validated, we also provide a theoretically justified way to choose $\gamma$ automatically for each sample and show it to outperform all the baseline models.

Informally, the intuition behind using focal loss is to direct the network's attention during training towards samples for which it is currently predicting a low probability value for the correct class, since trying to reduce the NLL on samples for which it is currently predicting a high probability value for the correct class is liable to lead to NLL overfitting, and thereby miscalibration (Guo* et al., 2017). More formally, we show in §4 that focal loss can be seen as implicitly regularising the weights of the network by causing the gradient norm to be lower than it would have been with cross-entropy loss as training proceeds, which we would theoretically expect to reduce overfitting and improve the calibration of the network. In §5, we perform extensive experiments on a variety of computer vision (CIFAR-10/100) and NLP (20 Newsgroups/SST) datasets, and with a wide variety of different network architectures (e.g. ResNet-110/50, Wide-ResNet, DenseNet), to show that this is indeed the case.

Our experiments show that in almost all cases, DNNs trained with focal loss are more calibrated than those trained with cross-entropy loss, MMCE, and Brier loss Brier (1950). Moreover, since our approach, like that of (Hendrycks et al., 2019), is complementary to the temperature scaling, significant improvements in calibration over temperature scaling alone, and state-of-the-art results, can be achieved by training with focal loss and then performing temperature scaling.

## 2 PROBLEM SETUP

Let $D = \langle (\mathbf{x}_i, y_i) \rangle_{i=1}^N$ denote a dataset consisting of samples from a joint distribution $\mathcal{D}(\mathcal{X}, \mathcal{Y})$, where for each sample $i$, $\mathbf{x}_i \in \mathcal{X}$ is the input and $y_i \in \mathcal{Y} = \{1, 2, ..., K\}$ is the ground-truth class label. Let $\hat{p}_{i,y} = f_\theta(y|\mathbf{x}_i)$ be the probability that a neural network $f$ with model parameters $\theta$ predicts for a class $y$ on a given input $\mathbf{x}_i$. The class that $f$ predicts for $\mathbf{x}_i$ is computed as $\hat{y}_i = \text{argmax}_{y \in \mathcal{Y}} \hat{p}_{i,y}$, and the predicted confidence as $\hat{p}_i = \max_{y \in \mathcal{Y}} \hat{p}_{i,y}$. Note that the confidence, by definition, does not depend on the ground-truth label. The network is said to be *perfectly calibrated* when, for each sample $(\mathbf{x}_i, y_i) \in D$, the confidence $\hat{p}_i$ is equal to the model accuracy $\mathbb{P}(\hat{y}_i = y_i)$, i.e. the probability that the predicted class is correct. For instance, of all the samples that a perfectly calibrated neural network classifies with a confidence of $0.8$, $80\%$ should be correctly predicted.

One of the most popular metrics used to measure model calibration is the *expected calibration error* (ECE) (Naeini et al., 2015), defined as the expected absolute difference between the model's confidence and its accuracy, i.e. $\mathbb{E}_{(\mathbf{x}_i, y_i) \sim \mathcal{D}} \big[ |\mathbb{P}(\hat{y}_i = y_i) - \hat{p}_i| \big]$. Since calculating $\mathbb{P}(\hat{y}_i = y_i)$ is infeasible, the ECE cannot be directly computed. A workaround is to divide the interval $[0, 1]$ into $M$ equispaced bins, where the $i^{\text{th}}$ bin is the interval $\left( \frac{i-1}{M}, \frac{i}{M} \right]$. Let $B_i$ denote the set of samples with confidence scores belonging to the $i^{\text{th}}$ bin. Then the accuracy $A_i$ of this bin is computed as $A_i = \frac{1}{|B_i|} \sum_{j \in B_i} \mathbb{1}(\hat{y}_j = y_j)$, where $\mathbb{1}$ is the indicator function, and $\hat{y}_j$ and $y_j$ are respectively the predicted and ground-truth labels for the $j^{\text{th}}$ sample. Similarly, the confidence $C_i$ of the $i^{\text{th}}$ bin is computed as $C_i = \frac{1}{|B_i|} \sum_{j \in B_i} \hat{p}_j$, i.e. $C_i$ is the average confidence of all samples in the bin. The ECE can be approximated as a weighted average of the absolute difference between the accuracy and confidence of each bin:

$$\text{ECE} = \sum_{i=1}^M \frac{|B_i|}{N} |A_i - C_i|. \tag{1}$$

A similar metric, the *maximum calibration error* (MCE) (Naeini et al., 2015), is defined as the maximum absolute difference between the confidence and accuracy of each bin:

$$\text{MCE} = \max_{i \in \{1,\ldots,M\}} |A_i - C_i|. \tag{2}$$

A common way of visually exploring the calibration of a model is to use a *reliability plot* (Niculescu-Mizil and Caruana, 2005), which plots the accuracies of the various confidence bins as a bar chart (see Figure 2). Reliability plots not only give a quick visual indication of a model's MCE (but not ECE, since the numbers of samples in the different bins are not shown), but also capture whether or not a model is *under-confident* or *over-confident* in general. For a perfectly calibrated model, the accuracy for each bin will match the confidence, and hence all of the bars will lie on a diagonal. By contrast, if most of the bars lie above the diagonal, meaning that the model is more accurate than it expects, then it is said to be under-confident, and if most of the bars lie below the diagonal, then it is said to be over-confident.

We also consider another metric which we call AdaECE (Adaptive ECE) where the bin sizes are decided such that each bin contains the same number of samples, unlike ECE (1) where every bin size is fixed and might contain very different number of samples (for example, Figure 2). However, AdaECE allows better approximation of ECE as it ensures that every approximate bin has the same sample density.

$$\text{AdaECE} = \sum_{i=1}^{M} \frac{|B_i|}{N} |A_i - C_i| \text{ where } \forall i \neq j, |B_i| = |B_j|. \tag{3}$$

## 3 WHAT CAUSES MISCALIBRATION?

We now discuss why high-capacity neural networks, despite achieving low classification errors on well-known datasets, tend to be miscalibrated. A key empirical observation made by (Guo* et al., 2017) was that poor calibration of such networks appears to be linked to overfitting on the negative log-likelihood (NLL) during training. In this section, we further inspect this observation to provide new insights, and discuss the main factors that play a role in causing the above-mentioned NLL overfitting.

For the analysis in this section, we train a ResNet-50 network on CIFAR-10 using a training infrastructure PyTorch-CIFAR which is known to produce state-of-the-art accuracy. We train it for 350 epochs using Stochastic Gradient Descent (SGD), with a momentum of 0.9, and a learning rate of 0.1 for the first 150 epochs, 0.01 for the next 100 epochs and 0.001 for the last 100 epochs. We use a mini-batch size of 128. Following standard practice, we minimise cross-entropy loss, which, for the $i^{th}$ training sample, is computed as $\mathcal{L}_c = -\log \hat{p}_{i,y_i}$ where $\hat{p}_{i,y_i}$ is the probability assigned by the network to the correct class for the $i^{th}$ sample.

It is interesting to note that NLL (a.k.a. cross-entropy) loss is a differentiable proxy for the actual metric that we aim to minimise by training a model, namely the classification error. However, there are certain differences between minimising the NLL and minimising the classification error. For instance, the NLL is minimised when for every training sample $i$, $\hat{p}_{i,y_i} = 1$, whereas the classification error is minimised if for every sample $i$, $\hat{p}_{i,y_i} > \hat{p}_{i,y}$ for all $y \neq y_i$. This indicates that even when the classification error is zero, the NLL can be positive, and the optimisation algorithm could in that case try to reduce it to zero by further increasing the value of $\hat{p}_{i,y_i}$ for each sample.

To study how miscalibration occurs during training, we plot the average NLL for the train and test sets at each training epoch in Figures 1(a) and 1(b). We also plot the average NLL and the entropy of the softmax distribution produced by the network for the correctly and incorrectly classified samples. In Figure 1(c), we plot the classification errors on the train and test sets, along with the test set ECE.

**Curse of misclassified samples:** Figures 1(a) and 1(b) show that although the average train NLL (for both correctly and incorrectly classified training samples) broadly decreases throughout training, after the $150^{th}$ epoch (where the learning rate drops by a factor of 10), there is a marked rise in the average test NLL, indicating that the network starts to overfit on average NLL. *However, the increase in average test NLL is caused only by the incorrectly classified samples*, as the average NLL for the correctly classified samples continues to decrease even after the $150^{th}$ epoch. We also observe that after epoch 150, the test set ECE rises, indicating that the network is becoming miscalibrated. This corroborates the observation in (Guo* et al., 2017) that miscalibration and NLL overfitting are linked.

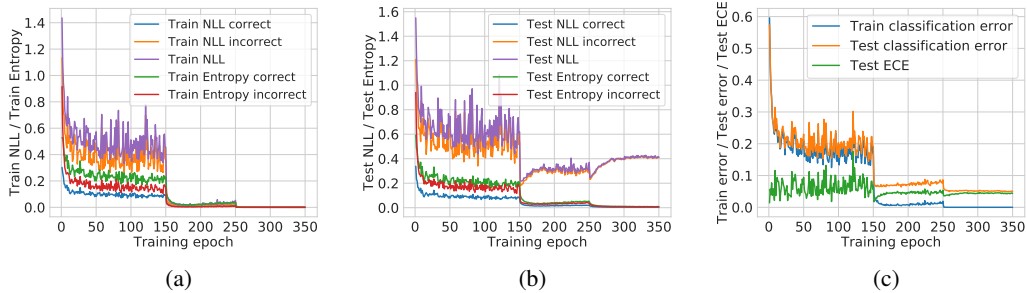

Figure 1: Metrics to capture how model calibration changes over training epochs in a ResNet-50 network trained on CIFAR-10.

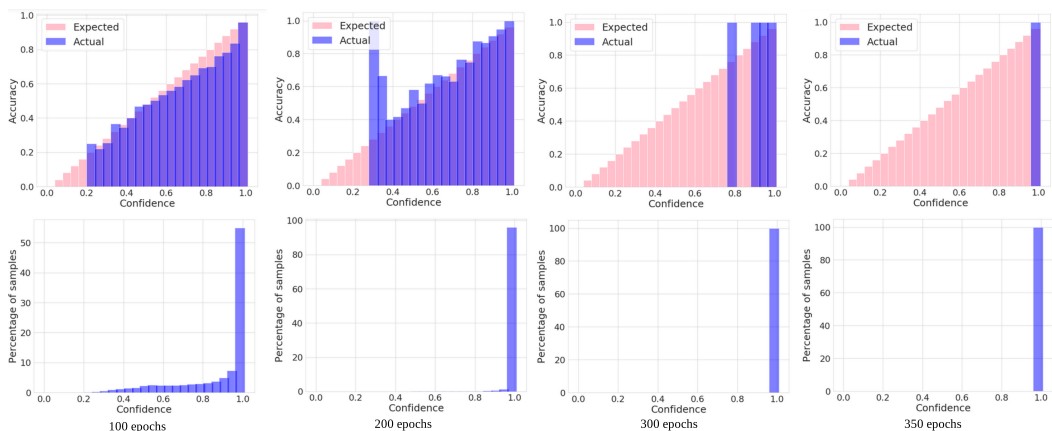

Figure 2: **Top row:** reliability plots using 25 confidence bins. **Bottom row:** % of samples in each bin, computed over the training set at epochs 100, 200, 300 and 350.

**Peak at the wrong place:** We further observe that the entropies of the softmax distributions for both the correctly and incorrectly classified test samples decrease throughout training (in other words, the distributions get peakier). This observation, coupled with the one we made above, indicates that for the wrongly classified test samples, the network gradually becomes more and more confident about its incorrect predictions. Finally, we notice that the training set classification error drops from around $20\%$ to nearly $0\%$ at epoch $150$, and further drops to exactly $0\%$ from epoch $250$ onwards. The network thus has a very low training set classification error throughout the period of miscalibration.

**Favouring correctly classified samples:** One potential explanation for the above observations is that after the $150^{th}$ training epoch, the network classifies almost all of the training samples correctly, and hence, in order to further minimise the training NLL, it focuses on increasing the confidence of its correct predictions, rather than focusing on the incorrectly classified samples. In order to experimentally validate this, we use two observations. Firstly, we divide the confidence range $[0, 1]$ into 25 bins, and present reliability plots computed on the training set at epochs 100, 200, 300 and 350 (see the top row of Figure 2). In Figure 2, we also show the percentage of samples in each confidence bin. It is quite clear from these plots that over time, the network gradually pushes all of the training samples towards the highest confidence bin. Furthermore, even though the network has achieved $100\%$ accuracy on the training set by epoch 300, it still pushes some of the samples lying in lower confidence bins to the highest confidence bin by epoch 350. *It thus keeps on increasing the confidence of correct predictions even after having reached perfect accuracy on the training set.* Secondly, we observe from Figure 1(c) that between training epochs $150$ and $250$, i.e. between the two points where there are drops in the learning rate, the training set classification error rises slightly, before dropping to exactly $0$ after epoch $250$. This further confirms our hypothesis that in this period, the network focuses more on increasing the confidence of its correct predictions rather than increasing the classification accuracy on the training set.

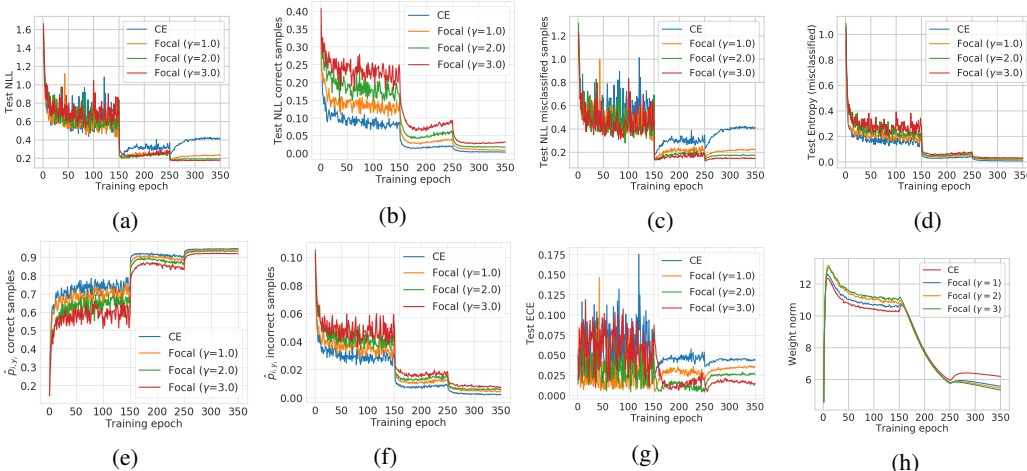

Figure 3: Metrics to compare the calibrations of several ResNet-50 networks trained on CIFAR-10, using either cross-entropy loss, or focal loss with $\gamma$ set to 1, 2 or 3.

**Weight magnification:** The above-mentioned increase in confidence of the network's predictions can happen if the network increases its weights to increase the magnitudes of the logits. We explore this in §4. This increase in the network's confidence during training is a key cause of miscalibration.

# 4 Improving Calibration using Focal Loss

As discussed above, overfitting on NLL, which is observed as the network grows more confident on all of its predictions irrespective of their correctness, is strongly related to poor calibration in neural networks. One cause of this phenomenon is that the cross-entropy objective minimises the difference between the softmax distribution and the ground-truth one-hot encoding over an entire mini-batch, irrespective of how well a network classifies individual samples in the mini-batch. In this work, we study an alternative loss function called *focal loss* (Lin et al., 2017), which tackles this problem by weighting loss components generated from individual samples in a mini-batch by how well the model classifies them. Focal loss for the $i^{th}$ sample in the training set is computed as $\mathcal{L}_f = -(1 - \hat{p}_{i,y_i})^\gamma \log \hat{p}_{i,y_i}$, where $\gamma \geq 0$ is a user-defined hyperparameter. Focal loss attributes more importance to samples that the network misclassifies than to ones that are correctly classified. In this paper, we study properties of focal loss that help in calibration.

**Empirical Observations:** In order to analyse the behaviour of neural networks trained on focal loss, we use the same framework as mentioned above, and train four ResNet-50 networks on CIFAR-10, one using cross-entropy loss, and three using focal loss with $\gamma = 1, 2$ or 3. We then observe various metrics throughout the training period of these networks, for comparison. In Figures 3(a), (b) and (c), we plot the average NLL over the test set, and for the correctly and incorrectly classified test samples, respectively, at each training epoch. In Figure 3(d), we plot the softmax entropy for the misclassified test samples. In Figures 3(e) and (f), we plot the average probability value $\hat{p}_{i,y_i}$ predicted for the ground-truth class, for correctly and incorrectly classified test samples, respectively. In Figure 3(g), we plot the test set ECE for all four models against the training epoch.

Figure 3(c) shows that in contrast to the network trained using cross-entropy, for which the NLL for misclassified test samples increases significantly after epoch 150, the rise in NLL for the networks trained on focal loss is much less severe. Moreover, from Figure 3(d), we notice that the softmax entropy for misclassified test samples is consistently (if marginally) higher for focal loss than for cross-entropy. *These observations indicate that the focal loss models are less confident about their incorrect predictions than the cross-entropy model.* From Figure 3(f), we notice that the probabilities predicted for the correct classes for misclassified test samples are slightly higher for focal loss than for cross-entropy. Conversely, Figure 3(e) shows that the probabilities predicted for the correct classes for correctly classified test samples are somewhat lower than for cross-entropy. (Note that in Figure 3(b), the average test NLL values for the focal loss networks on the correctly classified samples are also somewhat higher than for cross-entropy, which similarly indicates a slight decrease in confidence on those samples.) Nevertheless, for all four models, average confidence on correctly classified samples remains over 0.95 by the end of training, which is more than enough to make these

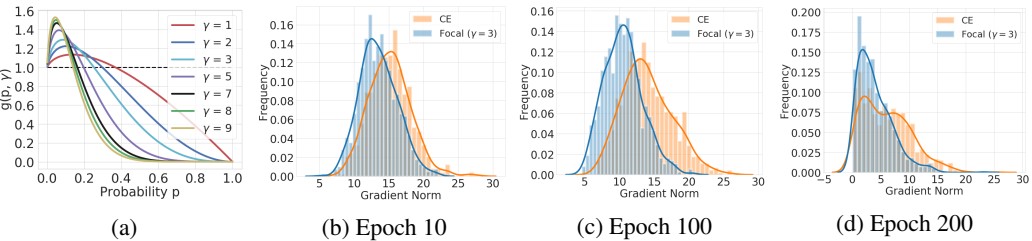

(a)         (b) Epoch 10         (c) Epoch 100         (d) Epoch 200

Figure 4: (a): $g(p, \gamma)$ vs $p$ and (b-d): histogram of the gradient norms of the last linear layer for both cross entropy and focal loss.

predictions usable by downstream tasks. Furthermore, the classification errors on the test set for all four models are also almost the same (refer fifth row of Table 3). Given both of these points, along with the ECE plot in Figue 3(g), it is thus reasonable to conclude that on a practical level, the models trained on focal loss are more calibrated, whilst retaining both their classification accuracy and the majority of their confidence on correct predictions.

**Theoretical Justification:** As mentioned previously, once a model trained using cross-entropy reaches high accuracy on the training set, the optimisation algorithm may try to further reduce the training NLL by increasing the confidence values for the correctly classified samples, rather than focusing on the misclassified samples. One way it could achieve this would be to increase the weights of the network to increase the magnitudes of the logits. In fact, this hypothesis would help to explain the observation made in (Guo* et al., 2017) that models trained using some form of weight decay are better calibrated.

To verify this hypothesis, we plot the $L_2$ norm of the weights of the last linear layer for all four networks as a function of the training epoch (see Figure 3(h)). It is interesting to note that although the norms of the weights for the models trained on focal loss are initially higher than that for the cross-entropy model, *a complete reversal* in the ordering of the weight norms occurs between epochs 150 and 250. In other words, as the network starts to become miscalibrated, the weight norm for the cross-entropy model also starts to become greater than those for the focal loss models. In practice, the reason for this is that focal loss, by design, starts to act as a regulariser on the network's weights once the model has gained a certain amount of confidence in its predictions. To better understand this, we start by considering the following proposition (see the Appendix B for a proof).

**Proposition 1.** *For focal loss $\mathcal{L}_f$ and cross-entropy loss $\mathcal{L}_c$, the gradients $\frac{\partial \mathcal{L}_f}{\partial \mathbf{w}} = \frac{\partial \mathcal{L}_c}{\partial \mathbf{w}} g(\hat{p}_{i,y_i}, \gamma)$, where $g(\hat{p}_{i,y_i}, \gamma) = (1 - \hat{p}_{i,y_i})^\gamma - \gamma \hat{p}_{i,y_i}(1 - \hat{p}_{i,y_i})^{\gamma-1} \log(\hat{p}_{i,y_i})$, $\gamma \geq 0$ is the focal loss hyperparameter, and $\mathbf{w}$ are the last layer network parameters. Thus, $\left\| \frac{\partial \mathcal{L}_f}{\partial \mathbf{w}} \right\| \leq \left\| \frac{\partial \mathcal{L}_c}{\partial \mathbf{w}} \right\|$ iff $g(\hat{p}_{i,y_i}, \gamma) \in [0, 1]$.*

To visualise the behaviour of $g(p, \gamma)$, we plot it against $p$ for various values of $\gamma$ in Figure 4(a). It is quite clear that $g(p, \gamma)$ lies in $[0, 1)$ for values of $p$ that lie above a certain threshold $p_0$. In the interval $[0, p_0)$, however, $g(p, \gamma) > 1$. In fact, for a given threshold $p_0$, we can efficiently compute (using the following proposition) a value $\gamma^*$ such that $g(p, \gamma^*) < 1$ for $p > p_0$ (see the Appendix B for a proof).

**Proposition 2.** *For $p \geq p_0$, $g(p, \gamma) \leq 1$ for all $\gamma \geq \gamma^* = \frac{a}{b} + \frac{1}{\log a} W_{-1}\left(-\frac{a^{(1-a/b)}}{b} \log a\right)$, where $a = 1 - p_0$, $b = p_0 \log p_0$ and $W_{-1}$ is the Lambert-W function (Corless et al., 1996). Moreover, for $p \geq p_0$ and $\gamma \geq \gamma^*$, the equality $g(p, \gamma) = 1$ holds only for $p = p_0$ and $\gamma = \gamma^*$.*

Thus, given a $p_0$, we can always use Proposition 2 to compute $\gamma^*$ such that (i) $g(p_0, \gamma^*) = 1$, (ii) for $p < p_0$, $g(p, \gamma^*) > 1$, and (iii) for $p > p_0$, $g(p, \gamma^*) < 1$.

**Implicit Regularisation:** For a network trained using focal loss with a fixed $\gamma$ obtained using $p_0$ such that $g(p_0, \gamma) = 1$ then in the initial stages of training, when $\hat{p}_{i,y_i} < p_0$ in general, $g(\hat{p}_{i,y_i}, \gamma) > 1$ and the model gains confidence on its predictions faster than it would for cross-entropy. However, as soon as $\hat{p}_{i,y_i}$ crosses the threshold $p_0$, $g(\hat{p}_{i,y_i}, \gamma)$ falls below 1 and reduces the size of the gradient updates made to the network weights, thereby having a regularising effect on the weights. This is why, in Figure 3(h), we find that the weight norms of the models trained with focal loss are initially higher than that for the model trained using cross-entropy. However, as training progresses, we find that the ordering of the weight norms reverses, as focal loss starts regularising the network weights.

Figure 4(b-d) give further insights by plotting histograms of the gradient norms of the last linear layer (over all samples in the training set) at epochs 10, 100 and 200, respectively. At epoch 10,

the gradient norms for cross-entropy and focal loss are similar, but as training progresses, those for cross-entropy decrease less rapidly than those for focal loss. Finally, note from Figure 4(a) that with higher values of $\gamma$, the fall in $g(\hat{p}_{i,y_i}, \gamma)$ gets steeper. We would thus expect the weight regularisation effect to be higher for models that use higher values of $\gamma$. This explains why, of the three models we trained using focal loss, the one with $\gamma = 3$ outperforms (in terms of calibration) the one with $\gamma = 2$, which in turn outperforms the model with $\gamma = 1$. Based on this observation, one might think that, in general, a higher value of gamma would lead to a more calibrated model. However, this is not the case, as we notice from Figure 4(a) that for $\gamma \geq 7$, $g(\hat{p}_{i,y_i}, \gamma)$ reduces to nearly 0 for a relatively low value of $\hat{p}_{i,y_i}$ (around 0.5). As a result, using values of $\gamma$ that are too high will cause the gradients to die (i.e. reduce to nearly 0) early, at a point at which the network's predictions remain ambiguous, thereby causing the training process to fail. *To this end, we provide Proposition 2 that allows us to efficiently get a sampled-dependent gamma in a principled manner.* Experimentally we show that the sample-dependant $\gamma$ does provide the best result among all the approaches we compare with. We also provide an empirical analysis of focal loss and cross entropy on a linear model in Appendix G.

## 5 EXPERIMENTS

We perform experiments on multiple image and document classification datasets to verify the effectiveness of focal loss for training calibrated models. For image classification experiments, we use CIFAR-10 (Krizhevsky, 2009) and CIFAR-100 (Krizhevsky, 2009) and for document classification we use 20 Newsgroups and the Stanford Sentiment Treebank (SST) (Socher et al., 2013). We provide details on the datasets and the train/validation/test splits for each dataset in the Appendix C. On CIFAR-10 and CIFAR-100, we train the networks: ResNet-50, ResNet-110 (He et al., 2016), Wide-ResNet-26-10 (Zagoruyko and Komodakis, 2016), and DenseNet-121 (Huang et al., 2017).We train the Global Pooling Convolutional Network (Lin et al., 2014) on 20 Newsgroups and the Tree-LSTM (Tai et al., 2015) on the SST Binary dataset. We provide implementation details for each of the state of the art networks in Appendix D. For each dataset-network pair, we train the network using each of the following loss functions:

**Baselines (Cross-Entropy, MMCE (Kumar et al., 2018)) and Brier Score (Brier, 1950)**: Models trained on cross-entropy loss, MMCE loss (i.e. cross-entropy with an MMCE regulariser) and Brier score serve as our baselines for comparison. MMCE (Maximum Mean Calibration Error) is a continuous and differentiable proxy for calibration error and hence, can be directly optimised using standard optimisers like Stochastic Gradient Descent. Normally, MMCE is optimised as a regulariser alongside cross-entropy. Brier score, for a single sample, is computed as the squared error between the predicted softmax vector and the one-hot ground truth encoding. We find Brier score to be a particularly relevant baseline for calibration as there is a distinct penalty for increasing the confidence on incorrect classes. In addition, it is relevant as it can be decomposed into calibration and refinement (DeGroot and Fienberg, 1983).

**Focal Loss (Fixed $\gamma$)**: We train various models each using three values of $\gamma$ set to 1, 2 and 3.

**Focal Loss (Sample-Dependent $\gamma$)**: From Proposition 2, we know that for a given probability $p$, we can find a value of $\gamma$ such that either $g(p, \gamma) \in [0, 1)$, and focal loss regularises the weights of the network, or $g(p, \gamma) > 1$, and focal loss speeds up the process of making the network more confident on the correct class. In this approach, we follow the idea of accelerating the rise in the value of $\hat{p}_{i,y_i}$ (the confidence on the correct class $y_i$ for the $i^{th}$ training sample) as long as $\hat{p}_{i,y_i} < 0.5$. As soon as $\hat{p}_{i,y_i} \geq 0.5$, we can confirm that the $i^{th}$ training sample has been predicted correctly. Hence, when $\hat{p}_{i,y_i} \geq 0.5$, we want to regularise the weights to prevent the network from becoming overconfident and miscalibrated. There can be multiple ways of choosing sample-dependent $\gamma$ that satisfy the above condition. However, with just few initial tries we found these two policies to produce quite competitive results on the validation sets: (a) Focal Loss (sample-wise $\gamma$ 5,3,2): $\gamma = 5$ for $\hat{p}_{i,y_i} \in [0, 0.19]$ (as $g(0.19, 5) \approx 1$), $\gamma = 3$ for $\hat{p}_{i,y_i} \in (0.19, 0.5]$ and $\gamma = 2$ for $\hat{p}_{i,y_i} \in (0.5, 1]$, and (b) Focal Loss (sample-wise $\gamma$ 5,3): $\gamma = 5$ for $\hat{p}_{i,y_i} \in [0, 0.19]$, and $\gamma = 3$ for $\hat{p}_{i,y_i} \in (0.19, 1]$.

**Focal Loss (Scheduled $\gamma$)**: As a simplification to the above approach, we also investigated the use of a schedule for $\gamma$ during training, as we expect the value of $\hat{p}_{i,y_i}$ to increase in general for all samples over time. In particular, we report results for two different schedules: (a) Focal Loss (scheduled $\gamma$ 5,3,2): $\gamma = 5$ for the first 100 epochs, $\gamma = 3$ for the next 150 epochs, and $\gamma = 2$ for the last 100 epochs, and (b) Focal Loss (scheduled $\gamma$ 5,3,1): $\gamma = 5$ for the first 100 epochs, $\gamma = 3$ for the next

| Dataset | Model | Cross Entropy | | Brier Loss | | MMCE | | Focal Loss (sample-wise $\gamma$ 5,3) | |
|---|---|---|---|---|---|---|---|---|---|
| | | Pre T | Post T | Pre T | Post T | Pre T | Post T | Pre T | Post T |
| CIFAR-100 | ResNet 50 | 17.52 | 3.42(2.10) | 6.52 | 3.64(1.10) | 15.32 | 2.38(1.80) | **4.50** | **2.00(1.10)** |
| | ResNet 110 | 19.05 | 4.43(2.30) | **7.88** | 4.65(1.20) | 19.14 | **3.86(2.30)** | 8.56 | 4.12(1.20) |
| | Wide ResNet 26-10 | 15.33 | 2.88(2.20) | 4.31 | 2.70(1.10) | 13.17 | 4.37(1.90) | **3.03** | **1.64(1.10)** |
| | DenseNet 121 | 20.98 | 4.27(2.30) | 5.17 | 2.29(1.10) | 19.13 | 3.06(2.10) | **3.73** | **1.31(1.10)** |
| CIFAR-10 | ResNet 50 | 4.35 | 1.35(2.50) | 1.82 | 1.08(1.10) | 4.56 | 1.19(2.60) | **1.55** | **0.95(1.10)** |
| | ResNet 110 | 4.41 | 1.09(2.80) | 2.56 | 1.25(1.20) | 5.08 | 1.42(2.80) | **1.87** | **1.07(1.10)** |
| | Wide ResNet 26-10 | 3.23 | 0.92(2.20) | **1.25** | 1.25(1.00) | 3.29 | 0.86(2.20) | 1.56 | **0.84(0.90)** |
| | DenseNet 121 | 4.52 | 1.31(2.40) | 1.53 | 1.53(1.00) | 5.10 | 1.61(2.50) | **1.22** | **1.22(1.00)** |
| 20 Newsgroups | Global Pooling CNN | 17.92 | 2.39(3.40) | 13.58 | 3.22(2.30) | 15.48 | 6.78(2.20) | **6.92** | **2.19(1.50)** |
| SST Binary | Tree LSTM | 7.37 | 2.62(1.80) | 9.01 | 2.79(2.50) | **5.03** | 4.02(1.50) | 9.19 | **1.83(0.70)** |

Table 1: ECE (%) computed for different approaches both pre and post temperature scaling (cross-validating T on ECE). Optimal temperature for each method is indicated in brackets.

| Dataset | Model | Cross Entropy | | Brier Loss | | MMCE | | Focal Loss (sample-wise $\gamma$ 5,3) | |
|---|---|---|---|---|---|---|---|---|---|
| | | Pre T | Post T | Pre T | Post T | Pre T | Post T | Pre T | Post T |
| CIFAR-100 | ResNet 50 | 17.52 | 3.67(2.10) | 6.52 | 3.69(1.20) | 15.32 | 2.44(1.80) | **4.39** | **2.33(1.10)** |
| | ResNet 110 | 19.05 | 5.50(2.40) | **7.73** | 4.53(1.20) | 19.14 | 4.85(2.30) | 8.55 | **3.96(1.20)** |
| | Wide ResNet 26-10 | 15.33 | 2.89(2.20) | 4.22 | 2.81(1.10) | 13.16 | 4.25(1.90) | **2.75** | **1.63(1.10)** |
| | DenseNet 121 | 20.98 | 5.09(2.30) | 5.04 | 2.56(1.10) | 19.13 | 3.07(2.10) | **3.55** | **1.24(1.10)** |
| CIFAR-10 | ResNet 50 | 4.33 | 2.14(2.50) | 1.74 | **1.23(1.10)** | 4.55 | 2.06(2.50) | **1.56** | 1.26(1.10) |
| | ResNet 110 | 4.40 | 1.84(2.70) | 2.60 | 1.70(1.20) | 5.06 | 2.45(2.70) | **2.07** | **1.67(1.10)** |
| | Wide ResNet 26-10 | 3.23 | 1.69(2.20) | 1.70 | 1.63(0.90) | 3.29 | 1.69(2.10) | **1.52** | **1.38(0.90)** |
| | DenseNet 121 | 4.51 | 2.25(2.30) | 2.03 | 2.18(0.90) | 5.10 | 2.46(2.40) | **1.42** | **1.42(1.00)** |
| 20 Newsgroups | Global Pooling CNN | 17.91 | 2.20(3.30) | 13.57 | 2.60(2.20) | 15.21 | 7.07(2.10) | **6.92** | **2.17(1.40)** |
| SST Binary | Tree LSTM | 7.31 | 2.10(1.80) | 9.42 | 2.80(2.70) | **4.49** | 3.72(1.50) | 8.63 | **1.92(0.70)** |

Table 2: Adaptive ECE (%) computed for different approaches both pre and post temperature scaling (cross-validating T on Adaptive ECE). Optimal temperature for each method is indicated in brackets.

150 epochs, and $\gamma = 1$ for the last 100 epochs. We also tried various other schedules, but found these two to produce the best results on the validation sets.

We find the best policy for each of the above three focal loss variants by cross-validating on the validation set, and report its results in Table 4 and 7. We report results for all other policies in Appendix E.

## 6 DISCUSSION

**Temperature Scaling:** We report the ECE % (using 15 bins), both before and after temperature scaling, for focal loss (sample-dependant which performed the best) vs baselines in Table 1. Further, we report the same metric for different variants of focal loss in Table 4. Similarly we report AdaECE in Table 2 and Table 5. In fact our results show that in situations where ECE might imply that the model is well calibrated, AdaECE brings out the miscalibration present in the model. For example, in the case of WideResNet on CIFAR10 for Cross Entropy, the best ECE obtained is 0.92, implying almost no miscalibration, whereas AdaECE is 1.69 showing further scope for improvements.

We compute the optimal temperature using two different approaches: (a) learning the optimal temperature by optimising NLL over a validation set, and (b) performing grid search over temperature values between 0 and 10, with a step of 0.1, and choosing the temperature that minimises the ECE/AdaECE over a validation set. We find the second approach to produce better results in general. Since we report ECE/AdaECE as the primary performance metric and grid search does not require a differentiable objective function, we directly minimise ECE/AdaECE over the validation set during grid search. Furthermore, since NLL is not a convex function of temperature, an optimisation algorithm may get stuck at a locally optimal temperature. We thus report the optimal temperatures and their corresponding ECEs/AdaECEs obtained using grid search (i.e. the second approach).

**Performance Gains:** It should be noted that for all the dataset-network pairs, we obtain state-of-the-art classification accuracies, as shown in Table 3. For focal loss with a fixed $\gamma$, we found that $\gamma = 3$ produced the best ECE results. This corroborates the observation we made in §4 that $\gamma = 3$ should produce better results than $\gamma = 1$ or $\gamma = 2$, as the regularising effect for $\gamma = 3$ is higher. For the sample-dependent $\gamma$ approach, we found the second policy (i.e. Focal Loss (sample-wise $\gamma$ 5,3) with $\gamma = 5$ for $\hat{p}_{i,y_i} \in [0, 0.19]$, and $\gamma = 3$ for $\hat{p}_{i,y_i} \in (0.19, 1]$) to produce better results. Of the two approaches for scheduled $\gamma$, we found the first schedule (i.e. Focal Loss (scheduled $\gamma$ 5,3,2) with

| Dataset | Model | Cross Entropy | Brier Loss | MMCE | FL-3 | FLS-532 | FLA-53 |
|---------|-------|---------------|------------|------|------|---------|--------|
| CIFAR-100 | ResNet 50 | 23.30 | 23.39 | 23.20 | 22.75 | 23.24 | 23.22 |
| | ResNet 110 | 22.73 | 25.10 | 23.07 | 22.92 | 22.96 | 22.51 |
| | Wide ResNet 26-10 | 20.70 | 20.59 | 20.73 | 19.69 | 20.13 | 20.11 |
| | DenseNet 121 | 24.52 | 23.75 | 24.00 | 23.25 | 23.72 | 22.67 |
| CIFAR-10 | ResNet 50 | 4.95 | 5.00 | 4.99 | 5.25 | 5.63 | 4.98 |
| | ResNet 110 | 4.89 | 5.48 | 5.40 | 5.08 | 5.71 | 5.42 |
| | Wide ResNet 26-10 | 3.86 | 4.08 | 3.91 | 4.13 | 4.46 | 4.01 |
| | DenseNet 121 | 5.00 | 5.11 | 5.41 | 5.33 | 5.65 | 5.46 |
| 20 Newsgroups | Global Pooling CNN | 26.68 | 27.06 | 27.23 | 29.26 | 28.16 | 27.98 |
| SST Binary | Tree LSTM | 12.85 | 12.85 | 11.86 | 12.19 | 13.07 | 12.80 |

Table 3: Error (%) computed for different approaches. In this table, FL-3 denotes Focal Loss (fixed $\gamma$ 3), FLS-532 denotes Focal Loss (scheduled $\gamma$ 5,3,2) and FLA-53 denotes Focal Loss (sample-wise $\gamma$ 5,3) i.e. focal loss with sample-wise $\gamma$ with $\gamma = 5$ for $\hat{p}_{i,y_i} \in [0, 0.19]$ and $\gamma = 3$ for $\hat{p}_{i,y_i} \in (0.19, 1]$.

| Dataset | Model | Focal Loss (fixed $\gamma$ 3) | | Focal Loss (scheduled $\gamma$ 5,3,2) | | Focal Loss (sample-wise $\gamma$ 5,3) | |
|---------|-------|------|------|------|------|------|------|
| | | Pre T | Post T | Pre T | Post T | Pre T | Post T |
| CIFAR-100 | ResNet 50 | 5.13 | **1.97(1.10)** | 8.47 | 2.13(1.30) | **4.50** | 2.00(1.10) |
| | ResNet 110 | 8.64 | 3.95(1.20) | 11.20 | **3.43(1.30)** | 8.56 | 4.12(1.20) |
| | Wide ResNet 26-10 | **2.13** | 2.13(1.00) | 4.98 | 1.94(1.20) | 3.03 | **1.64(1.10)** |
| | DenseNet 121 | 4.15 | **1.25(1.10)** | 7.63 | 1.96(1.20) | **3.73** | 1.31(1.10) |
| CIFAR-10 | ResNet 50 | **1.48** | 1.42(1.10) | 2.97 | 1.53(1.20) | 1.55 | **0.95(1.10)** |
| | ResNet 110 | **1.55** | **1.02(1.10)** | 3.33 | 1.36(1.30) | 1.87 | 1.07(1.10) |
| | Wide ResNet 26-10 | 1.69 | 0.97(0.90) | 1.82 | 1.45(1.10) | **1.56** | **0.84(0.90)** |
| | DenseNet 121 | 1.32 | 1.26(0.90) | 2.22 | 1.34(1.10) | **1.22** | 1.22(1.00) |
| 20 Newsgroups | Global Pooling CNN | 8.67 | 3.51(1.50) | 12.13 | 2.47(1.80) | **6.92** | **2.19(1.50)** |
| SST Binary | Tree LSTM | 16.05 | **1.78(0.50)** | **3.91** | 2.64(0.90) | 9.19 | 1.83(0.70) |

Table 4: ECE (%) computed for different approaches both pre and post temperature scaling (cross-validating T on ECE). Optimal temperature for each method is indicated in brackets.

| Dataset | Model | Focal Loss (fixed $\gamma$ 3) | | Focal Loss (scheduled $\gamma$ 5,3,2) | | Focal Loss (sample-wise $\gamma$ 5,3) | |
|---------|-------|------|------|------|------|------|------|
| | | Pre T | Post T | Pre T | Post T | Pre T | Post T |
| CIFAR-100 | ResNet 50 | 5.08 | 2.35(1.20) | 8.41 | **2.25(1.30)** | **4.39** | 2.33(1.10) |
| | ResNet 110 | 8.64 | 4.14(1.20) | 11.18 | **3.68(1.30)** | 8.55 | 3.96(1.20) |
| | Wide ResNet 26-10 | **2.08** | 2.08(1.00) | 5.00 | 2.11(1.20) | 2.75 | **1.63(1.10)** |
| | DenseNet 121 | 4.15 | **1.23(1.10)** | 7.61 | 2.04(1.20) | **3.55** | 1.24(1.10) |
| CIFAR-10 | ResNet 50 | 1.95 | 1.83(1.10) | 2.95 | 2.18(1.20) | **1.56** | **1.26(1.10)** |
| | ResNet 110 | **1.62** | **1.44(1.10)** | 3.32 | 1.91(1.40) | 2.07 | 1.67(1.10) |
| | Wide ResNet 26-10 | 1.84 | 1.54(0.90) | 2.04 | 1.90(1.10) | **1.52** | **1.38(0.90)** |
| | DenseNet 121 | **1.22** | 1.48(0.90) | 2.19 | 1.59(1.20) | 1.42 | **1.42(1.00)** |
| 20 Newsgroups | Global Pooling CNN | 8.65 | 3.78(1.50) | 12.13 | 2.18(1.90) | **6.92** | **2.17(1.40)** |
| SST Binary | Tree LSTM | 15.64 | 2.17(0.50) | **2.94** | 2.50(0.90) | 8.63 | **1.92(0.70)** |

Table 5: Adaptive ECE (%) computed for different approaches both pre and post temperature scaling (cross-validating T on Adaptive ECE). Optimal temperature for each method is indicated in brackets.

$\gamma = 5$ for the first 100 epochs, $\gamma = 3$ for the next 150 epochs, and $\gamma = 2$ for the last 100 epochs) to produce better ECEs/AdaECEs for every model and dataset. *We report the ECE values for the best performing approaches in Table 1 and 4.* Similarly we show the AdaECE values of these models in 2 and 5. Full results (ECE, AdaECE, MCE, NLL and test error) for all approaches are reported in the Appendix E.

It is clear from Table 1 and 4 that focal loss with sample-dependent $\gamma$ outperforms all the baselines (cross-entropy, brier score and MMCE). It broadly produces the lowest ECE and AdaECE values both before and after temperature scaling. This observation is particularly encouraging, as it indicates that a principled method of obtaining values of $\gamma$ for focal loss can work well, thereby *alleviating the need to naively cross-validate the $\gamma$ hyperparameter*. Also, Table 4 and Table 5 show that other focal loss based approaches are also very competitive.

| Dataset | Model | Cross Entropy (Pre T) | | Cross Entropy (Post T) | | MMCE (Pre T) | | MMCE (Post T) | | Focal Loss (Pre T) | | Focal Loss (Post T) | |
|---|---|---|---|---|---|---|---|---|---|---|---|---|---|
| | | |S99|% | Accuracy | |S99|% | Accuracy | |S99|% | Accuracy | |S99|% | Accuracy | |S99|% | Accuracy | |S99|% | Accuracy |
| CIFAR-10 | ResNet 110 | 97.11 | 96.33 | 11.5 | 97.39 | 97.65 | 96.72 | 10.62 | 99.83 | 61.41 | 99.51 | 31.10 | 99.68 |
| CIFAR-10 | ResNet 50 | 95.93 | 96.72 | 7.33 | 99.73 | 92.33 | 98.24 | 4.21 | 100 | 46.31 | 99.57 | 14.27 | 99.93 |

Table 6: Percentage of test samples predicted with confidence higher than $99\%$ and the corresponding accuracy for Cross Entropy, MMCE and Focal loss computed both pre and post temperature scaling (represented in the table as pre T and post T respectively).

**Confident and Calibrated Models:** It is interesting to note that for focal loss with sample-based $\gamma$ (refer Tab. 1 and Tab. 2), beside most other focal loss models, the optimal temperatures are very close to 1, mostly lying between 0.9 and 1.1. By contrast, the optimal temperatures for the baselines (cross-entropy and MMCE) are significantly higher, with values lying between 2.0 to 2.8. An optimal temperature close to 1 indicates that the model is innately calibrated and cannot be made significantly more calibrated by temperature scaling. Furthermore, an optimal temperature that is much greater than 1 can make the network underconfident in general, as its outputs are temperature-scaled irrespective of their correctness. To see this, we follow the approach adopted in Kumar et al. (2018), and measure the percentage of test samples that are predicted with a confidence of 0.99 or more (we call this set of test samples $S99$). In Table 6, we report $|S99|$ as a percentage of the total number of test samples, along with the accuracy of the samples in $S99$ for ResNet-50 and ResNet-110 trained on CIFAR-10, using cross-entropy loss, MMCE loss, and focal loss. We observe that $|S99|$ for the focal loss model is much lower than for the cross-entropy or MMCE models before temperature scaling. However, after temperature scaling, $|S99|$ for focal loss is significantly higher than for both MMCE and cross-entropy. The reason is that with an optimal temperature of 1.1, the confidence of the temperature-scaled model for focal loss does not reduce as much as those of the models for cross-entropy and MMCE, for which the optimal temperatures lie between 2.5 to 2.8. We thus conclude that models trained on focal loss are not only more calibrated, but also better preserve their confidence on predictions, even after being post-processed with temperature scaling. We present some qualitative results to support this claim in Appendix F.

## 7 Conclusion

In this paper, we have shown that training using focal loss can yield multi-class classification networks that are more naturally calibrated than those trained using more conventional cross-entropy loss. There are sound theoretical reasons to expect this: in particular, as we show in §4, focal loss implicitly regularises the weights of a network during training, reducing NLL overfitting and thereby improving calibration. Extensive experiments on a variety of computer vision (CIFAR-10/100) and NLP (20 Newsgroups/SST) datasets, and with a wide variety of different network architectures, show that this expectation is also borne out in practice. Our results show that in almost all cases, networks trained with focal loss are more calibrated than those trained with cross-entropy loss, whilst having similar levels of accuracy, making their predictions much easier for downstream components to trust. We also introduce principled way of getting sample-dependent $\gamma$ for focal loss and we show that it produces good results across datasets and models.

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

## A  APPENDIX

In § B, we show the proofs of the propositions formulated in the main text. We then describe all the datasets we used for our experiments in § C which is followed by implementation details of the state of the art networks in § D.

## B  PROOFS

Here we provide the proofs of both the propositions presented in the main text. While Proposition 3 helps us understand the regularization effect of using the focal loss, the Proposition 4 efficiently provides us the $\gamma$ values in a principle way such that it is sample-dependent. Implementing the sample-dependent $\gamma$ is very easy as implementation of the Lambert-W function (Corless et al., 1996) is available in standard libraries (e.g. python scipy). Using it we could get quite competitive results with just few initial sample-dependent $\gamma$ policies that we tried.

**Proposition 3.** *For focal loss $\mathcal{L}_f$ and cross-entropy loss $\mathcal{L}_c$, the gradients $\frac{\partial \mathcal{L}_f}{\partial \mathbf{w}} = \frac{\partial \mathcal{L}_c}{\partial \mathbf{w}} g(\hat{p}_{i,y_i}, \gamma)$, where $g(\hat{p}_{i,y_i}, \gamma) = (1 - \hat{p}_{i,y_i})^\gamma - \gamma \hat{p}_{i,y_i}(1 - \hat{p}_{i,y_i})^{\gamma-1} \log(\hat{p}_{i,y_i})$, $\gamma \geq 0$ is the focal loss hyperparameter, and $\mathbf{w}$ are the last layer network parameters. Thus, $\left\| \frac{\partial \mathcal{L}_f}{\partial \mathbf{w}} \right\| \leq \left\| \frac{\partial \mathcal{L}_c}{\partial \mathbf{w}} \right\|$ iff $g(\hat{p}_{i,y_i}, \gamma) \in [0, 1]$.*

*Proof.* Let $\mathbf{w}$ be the linear layer parameters connecting the feature map to the logit $s$. Then, using the chain rule, $\frac{\partial \mathcal{L}_f}{\partial \mathbf{w}} = \left(\frac{\partial s}{\partial \mathbf{w}}\right)\left(\frac{\partial \hat{p}_{i,y_i}}{\partial s}\right)\left(\frac{\partial \mathcal{L}_f}{\partial \hat{p}_{i,y_i}}\right)$. Similarly, $\frac{\partial \mathcal{L}_c}{\partial \mathbf{w}} = \left(\frac{\partial s}{\partial \mathbf{w}}\right)\left(\frac{\partial \hat{p}_{i,y_i}}{\partial s}\right)\left(\frac{\partial \mathcal{L}_c}{\partial \hat{p}_{i,y_i}}\right)$. The derivative of the focal loss with respect to $\hat{p}_{i,y_i}$, the softmax output of the network for the true class $y_i$, takes the form

$$\frac{\partial \mathcal{L}_f}{\partial \hat{p}_{i,y_i}} = -\frac{1}{\hat{p}_{i,y_i}}\left((1 - \hat{p}_{i,y_i})^\gamma - \gamma \hat{p}_{i,y_i}(1 - \hat{p}_{i,y_i})^{\gamma-1} \log(\hat{p}_{i,y_i})\right)$$
$$= \frac{\partial \mathcal{L}_c}{\partial \hat{p}_{i,y_i}} g(\hat{p}_{i,y_i}, \gamma), \tag{4}$$

in which $g(\hat{p}_{i,y_i}, \gamma) = (1 - \hat{p}_{i,y_i})^\gamma - \gamma \hat{p}_{i,y_i}(1 - \hat{p}_{i,y_i})^{\gamma-1} \log(\hat{p}_{i,y_i})$ and $\frac{\partial \mathcal{L}_c}{\partial \hat{p}_{i,y_i}} = -\frac{1}{\hat{p}_{i,y_i}}$. It is thus straightforward to verify that if $g(\hat{p}_{i,y_i}, \gamma) \in [0, 1]$, then $\left\| \frac{\partial \mathcal{L}_f}{\partial \hat{p}_{i,y_i}} \right\| \leq \left\| \frac{\partial \mathcal{L}_c}{\partial \hat{p}_{i,y_i}} \right\|$, which itself implies that $\left\| \frac{\partial \mathcal{L}_f}{\partial \mathbf{w}} \right\| \leq \left\| \frac{\partial \mathcal{L}_c}{\partial \mathbf{w}} \right\|$. □

**Proposition 4.** *For $p \geq p_0$, $g(p, \gamma) \leq 1$ for all $\gamma \geq \gamma^* = \frac{a}{b} + \frac{1}{\log a} W_{-1}\left(-\frac{a^{(1-a/b)}}{b} \log a\right)$, where $a = 1 - p_0$, $b = p_0 \log p_0$ and $W_{-1}$ is the Lambert-W function (Corless et al., 1996). Moreover, for $p \geq p_0$ and $\gamma \geq \gamma^*$, the equality $g(p, \gamma) = 1$ holds only for $p = p_0$ and $\gamma = \gamma^*$.*

*Proof.* We derive the value of $\gamma > 0$ for which $g(p_0, \gamma) = 1$ for a given $p_0 \in [0, 1]$. From Proposition 4.1, we already know that

$$\frac{\partial \mathcal{L}_f}{\partial \hat{p}_{i,y_i}} = \frac{\partial \mathcal{L}_c}{\partial \hat{p}_{i,y_i}} g(\hat{p}_{i,y_i}, \gamma), \tag{5}$$

where $\mathcal{L}_f$ is focal loss, $\mathcal{L}_c$ is cross entropy loss, $\hat{p}_{i,y_i}$ is the probability assigned by the model to the ground-truth correct class for the $i^{th}$ sample, and

$$g(\hat{p}_{i,y_i}, \gamma) = (1 - \hat{p}_{i,y_i})^\gamma - \gamma \hat{p}_{i,y_i}(1 - \hat{p}_{i,y_i})^{\gamma-1} \log(\hat{p}_{i,y_i}). \tag{6}$$

For $p \in [0, 1]$, if we look at the function $g(p, \gamma)$, then we can clearly see that $g(p, \gamma) \to 1$ as $p \to 0$, and that $g(p, \gamma) = 0$ when $p = 1$. To observe the behaviour of $g(p, \gamma)$ for intermediate values of $p$, we first take its derivative with respect to $p$:

$$\frac{\partial g(p, \gamma)}{\partial p} = \gamma(1 - p)^{\gamma-2}\left[-2(1 - p) - (1 - p)\log p + (\gamma - 1)p \log p\right] \tag{7}$$

In Equation 7, $\gamma(1 - p)^{\gamma-2} > 0$ except when $p = 1$ (in which case $\gamma(1 - p)^{\gamma-2} = 0$). Thus, to observe the sign of the gradient $\frac{\partial g(p, \gamma)}{\partial p}$, we focus on the term

$$-2(1 - p) - (1 - p)\log p + (\gamma - 1)p \log p. \tag{8}$$

Dividing Equation 8 by $(-\log p)$, the sign remains unchanged and we get

$$k(p, \gamma) = \frac{2(1-p)}{\log p} + 1 - \gamma p. \tag{9}$$

We can see that $k(p, \gamma) \to 1$ as $p \to 0$ and $k(p, \gamma) \to -(1+\gamma)$ as $p \to 1$ (using l'Hôpital's rule). Furthermore, $k(p, \gamma)$ is monotonically decreasing for $p \in [0, 1]$. Thus, as the gradient $\frac{\partial g(p,\gamma)}{\partial p}$ monotonically decreases from a positive value at $p = 0$ to a negative value at $p = 1$, we can say that $g(p, \gamma)$ first monotonically increases starting from 1 (as $p \to 0$) and then monotonically decreases down to 0 (at $p = 1$). Thus, if for some threshold $p_0 > 0$ and for some $\gamma > 0$, $g(p, \gamma) = 1$, then $\forall p > p_0, g(p, \gamma) < 1$. We now want to find a $\gamma$ such that $\forall p \geq p_0, g(p, \gamma) \leq 1$. First, let $a = (1-p_0)$ and $b = p_0 \log p_0$. Then:

$$\begin{aligned}
g(p_0, \gamma) &= (1-p_0)^\gamma - \gamma p_0 (1-p_0)^{\gamma-1} \log p_0 \leq 1 \\
&\implies (1-p_0)^{\gamma-1}[(1-p_0) - \gamma p_0 \log p_0] \leq 1 \\
&\implies a^{\gamma-1}(a - \gamma b) \leq 1 \\
&\implies (\gamma - 1)\log a + \log(a - \gamma b) \leq 0 \\
&\implies \left(\gamma - \frac{a}{b}\right)\log a + \log(a - \gamma b) \leq \left(1 - \frac{a}{b}\right)\log a \\
&\implies (a - \gamma b)e^{(\gamma - a/b)\log a} \leq a^{(1-a/b)} \\
&\implies \left(\gamma - \frac{a}{b}\right)e^{(\gamma - a/b)\log a} \leq -\frac{a^{(1-a/b)}}{b} \\
&\implies \left(\left(\gamma - \frac{a}{b}\right)\log a\right)e^{(\gamma - a/b)\log a} \geq -\frac{a^{(1-a/b)}}{b}\log a
\end{aligned} \tag{10}$$

where $a = (1-p_0)$ and $b = p_0 \log p_0$. We know that the inverse of $y = xe^x$ is defined as $x = W(y)$, where $W$ is the Lambert-W function (Corless et al., 1996). Furthermore, the r.h.s. of the inequality in Equation 10 is always negative, with a minimum possible value of $-1/e$ that occurs at $p_0 = 0.5$. Therefore, applying the Lambert-W function to the r.h.s. will yield two real solutions (corresponding to a principal branch denoted by $W_0$ and a negative branch denoted by $W_{-1}$). We first consider the solution corresponding to the negative branch (which is the smaller of the two solutions):

$$\begin{aligned}
\left((\gamma - \frac{a}{b})\log a\right) &\leq W_{-1}\left(-\frac{a^{(1-a/b)}}{b}\log a\right) \\
&\implies \gamma \geq \frac{a}{b} + \frac{1}{\log a}W_{-1}\left(-\frac{a^{(1-a/b)}}{b}\log a\right)
\end{aligned} \tag{11}$$

If we consider the principal branch, the solution is

$$\gamma \leq \frac{a}{b} + \frac{1}{\log a}W_0\left(-\frac{a^{(1-a/b)}}{b}\log a\right), \tag{12}$$

which yields a negative value for $\gamma$ that we discard. Thus Equation 11 gives the values of $\gamma$ for which if $p > p_0$, then $g(p, \gamma) < 1$. In other words, $g(p_0, \gamma) = 1$, and for any $p < p_0, g(p, \gamma) > 1$. □

## C  DATASET DESCRIPTION

We use the following image- and document-classification datasets in our experiments:

1. **CIFAR-10** (Krizhevsky, 2009): This dataset has 60,000 colour images of size $32 \times 32$, divided equally into 10 classes. We use a train/validation/test split of 45,000/5,000/10,000 images.

2. **CIFAR-100** (Krizhevsky, 2009): This dataset has 60,000 colour images of size $32 \times 32$, divided equally into 100 classes. (Note that the images in this dataset are not the same images as in CIFAR-10.) We again use a train/validation/test split of 45,000/5,000/10,000 images.

3. **20 Newsgroups**: This dataset contains 20,000 news articles, categorised evenly into 20 different newsgroups based on their content. It is a popular dataset for text classification. Whilst some of the newsgroups are very related (e.g. rec.motorcycles and rec.autos), others are quite unrelated (e.g. sci.space and misc.forsale). We use a train/validation/test split of 15,098/900/3,999 documents.

4. **Stanford Sentiment Treebank (SST)** (Socher et al., 2013): This dataset contains movie reviews in the form of sentence parse trees, where each node is annotated by sentiment. We use the dataset version with binary labels, for which 6,920/872/1,821 documents are used as the training/validation/test split. In the training set, each node of a parse tree is annotated as positive, neutral or negative. At test time, the evaluation is done based on the model classification at the root node, i.e. considering the whole sentence, which contains only positive or negative sentiment.

## D    IMPLEMENTATION DETAILS

For training networks on the image classification datasets (CIFAR-10 and CIFAR-100), we use SGD with a momentum of 0.9 as our optimiser, and train the networks for 350 epochs, with a learning rate of 0.1 for the first 150 epochs, 0.01 for the next 100 epochs, and 0.001 for the last 100 epochs. We use a training batch size of 128. Furthermore, we augment the training images by applying random crops and random horizontal flips.

For 20 Newsgroups, we train the Global Pooling Convolutional Network (Lin et al., 2014) using the Adam optimiser, with learning rate 0.001, and betas 0.9 and 0.999. The code is a PyTorch adaptation of (Ng). We used Glove word embeddings (Pennington et al., 2014) to train the network. We trained all the models for 50 epochs and used the models with the best validation accuracy.

For the SST Binary dataset, we train the Tree-LSTM (Tai et al., 2015) using the AdaGrad optimiser with a learning rate of 0.05 and a weight decay of $10^{-4}$, as suggested by the authors. We used the constituency model, which considers binary parse trees of the data and trains a binary tree LSTM on them. The Glove word embeddings (Pennington et al., 2014) we used were also tuned for best results. The code framework we used is inspired by (TreeLSTM). We trained all of our models for 25 epochs and used the models with the best validation accuracy.

For all our models, we use the PyTorch framework, setting any hyperparameters not explicitly mentioned to the default values used in the standard models. For MMCE, we used $\lambda = 2$ for all the image-classification tasks, whilst we found $\lambda = 8$ to perform better for document classification. A calibrated model which does not generalise well to an unseen test set is not very useful. Hence, for all the experiments, we set the training parameters in a way such that we get state-of-the-art test set accuracies on all datasets for each model.

## E    ADDITIONAL RESULTS

We use various metrics to compare the proposed methods based on focal loss with the baselines (i.e. cross-entropy, Brier score and MMCE).

We show all test ECE of all the focal loss models before and after temperature scaling in Table 7. Similarly, we show all test AdaECE of all the focal loss models before and after temperature scaling in Table 8. We present the test set error for all the focal loss models in Table 9. We present the test NLL % before and after temperature scaling in Tables 10 and 11, respectively. We report the test set MCE % before and after temperature scaling in Tables 12 and 13, respectively.

In this section we use the following abbreviation to report results on different varieties of Focal Loss. FL-1 refers to Focal Loss (fixed $\gamma$ 1), FL-2 refers to Focal Loss (fixed $\gamma$ 2), FL-3 refers to Focal Loss (fixed $\gamma$ 3), FLS-531 refers to Focal Loss (scheduled $\gamma$ 5,3,1), FLS-532 refers to Focal Loss (scheduled $\gamma$ 5,3,2), FLA-532 refers to Focal Loss (sample-wise $\gamma$ 5,3,2) and FLA-53 refers to Focal Loss (sample-wise $\gamma$ 5,3)

## F    QUALITATIVE RESULTS

In Figure 8, we present some qualitative results to show the improvement in the confidence estimates of focal loss in comparison to other baselines (i.e., cross entropy, MMCE and Brier score). For this, we take ResNet-50 networks trained on CIFAR-10 using all the four loss functions (cross entropy, MMCE, Brier score and Focal loss with sample-wise $\gamma$ 5,3) and measure the confidence of their predictions for four correctly and four incorrectly classified test samples. We report these confidences both before and after temperature scaling. It is clear from Figure 8 that for all the

| Dataset | Model | FL-1 | | FL-2 | | FL-3 | | FLS-531 | | FLS-532 | | FLA-532 | | FLA-53 | |
|---|---|---|---|---|---|---|---|---|---|---|---|---|---|---|---|
| | | Pre T | Post T | Pre T | Post T | Pre T | Post T | Pre T | Post T | Pre T | Post T | Pre T | Post T | Pre T | Post T |
| CIFAR-100 | ResNet 50 | 12.86 | 2.29(1.50) | 8.61 | 2.24(1.30) | 5.13 | 1.97(1.10) | 11.63 | 2.09(1.40) | 8.47 | 2.13(1.30) | 9.09 | 1.61(1.30) | 4.50 | 2.00(1.10) |
| | ResNet 110 | 15.08 | 4.55(1.50) | 11.56 | 3.72(1.30) | 8.64 | 3.95(1.20) | 14.99 | 4.56(1.50) | 11.20 | 3.43(1.30) | 11.74 | 3.64(1.30) | 8.56 | 4.12(1.20) |
| | Wide ResNet 26-10 | 8.93 | 2.53(1.40) | 4.64 | 2.93(1.20) | 2.13 | 2.13(1.00) | 9.36 | 2.48(1.40) | 4.98 | 1.94(1.20) | 4.98 | 2.55(1.20) | 3.03 | 1.64(1.10) |
| | DenseNet 121 | 14.24 | 2.80(1.50) | 7.90 | 2.33(1.20) | 4.15 | 1.25(1.10) | 13.05 | 2.08(1.50) | 7.63 | 1.96(1.20) | 8.14 | 2.35(1.30) | 3.73 | 1.31(1.10) |
| CIFAR-10 | ResNet 50 | 3.42 | 1.08(1.60) | 2.36 | 0.91(1.20) | 1.48 | 1.42(1.10) | 4.06 | 1.53(1.60) | 2.97 | 1.53(1.20) | 2.52 | 0.88(1.30) | 1.55 | 0.95(1.10) |
| | ResNet 110 | 3.46 | 1.20(1.60) | 2.70 | 0.89(1.30) | 1.55 | 1.02(1.10) | 4.92 | 1.50(1.70) | 3.33 | 1.36(1.30) | 2.82 | 0.97(1.30) | 1.87 | 1.07(1.10) |
| | Wide ResNet 26-10 | 2.69 | 1.46(1.30) | 1.42 | 1.03(1.10) | 1.69 | 0.97(0.90) | 2.81 | 0.96(1.40) | 1.82 | 1.45(1.10) | 1.31 | 0.87(1.10) | 1.56 | 0.84(0.90) |
| | DenseNet 121 | 3.44 | 1.63(1.40) | 1.93 | 1.04(1.10) | 1.32 | 1.26(0.90) | 4.12 | 1.65(1.50) | 2.22 | 1.34(1.10) | 2.45 | 1.31(1.20) | 1.22 | 1.22(1.00) |
| 20 Newsgroups | Global Pooling CNN | 15.06 | 2.14(2.60) | 12.10 | 3.22(1.60) | 8.67 | 3.51(1.50) | 13.55 | 4.32(1.70) | 12.13 | 2.47(1.80) | 12.20 | 2.39(2.00) | 6.92 | 2.19(1.50) |
| SST Binary | Tree LSTM | 6.78 | 3.29(1.60) | 3.05 | 3.05(1.00) | 16.05 | 1.78(0.50) | 4.66 | 3.36(1.40) | 3.91 | 2.64(0.90) | 4.47 | 2.77(0.90) | 9.19 | 1.83(0.70) |

Table 7: ECE (%) computed for different focal loss approaches both pre and post temperature scaling (cross-validating T on ECE). Optimal temperature for each method is indicated in brackets.

| Dataset | Model | FL-1 | | FL-2 | | FL-3 | | FLS-531 | | FLS-532 | | FLA-532 | | FLA-53 | |
|---|---|---|---|---|---|---|---|---|---|---|---|---|---|---|---|
| | | Pre T | Post T | Pre T | Post T | Pre T | Post T | Pre T | Post T | Pre T | Post T | Pre T | Post T | Pre T | Post T |
| CIFAR-100 | ResNet 50 | 12.86 | 2.54(1.50) | 8.55 | 2.44(1.30) | 5.08 | 2.35(1.20) | 11.58 | 2.01(1.40) | 8.41 | 2.25(1.30) | 9.08 | 1.94(1.30) | 4.39 | 2.33(1.10) |
| | ResNet 110 | 15.08 | 4.30(1.60) | 11.57 | 4.38(1.40) | 8.64 | 4.14(1.20) | 14.98 | 3.92(1.60) | 11.18 | 3.68(1.30) | 11.74 | 4.21(1.30) | 8.55 | 3.96(1.20) |
| | Wide ResNet 26-10 | 8.93 | 2.74(1.40) | 4.65 | 2.96(1.20) | 2.08 | 2.08(1.00) | 9.20 | 2.52(1.40) | 5.00 | 2.11(1.20) | 5.00 | 2.58(1.20) | 2.75 | 1.63(1.10) |
| | DenseNet 121 | 14.24 | 2.71(1.50) | 7.90 | 2.36(1.20) | 4.15 | 1.23(1.10) | 13.01 | 2.18(1.50) | 7.61 | 2.04(1.20) | 8.04 | 2.10(1.30) | 3.55 | 1.24(1.10) |
| CIFAR-10 | ResNet 50 | 3.42 | 1.51(1.60) | 2.37 | 1.69(1.20) | 1.95 | 1.83(1.10) | 4.06 | 2.36(1.50) | 2.95 | 2.18(1.20) | 2.50 | 1.23(1.30) | 1.56 | 1.26(1.10) |
| | ResNet 110 | 3.42 | 1.57(1.70) | 2.69 | 1.29(1.30) | 1.62 | 1.44(1.10) | 4.91 | 2.62(1.60) | 3.32 | 1.91(1.40) | 2.78 | 1.58(1.30) | 2.07 | 1.67(1.10) |
| | Wide ResNet 26-10 | 2.70 | 1.52(1.40) | 1.64 | 1.47(1.10) | 1.84 | 1.54(0.90) | 2.75 | 1.87(1.30) | 2.04 | 1.90(1.10) | 1.68 | 1.49(1.10) | 1.52 | 1.38(0.90) |
| | DenseNet 121 | 3.44 | 1.85(1.40) | 1.80 | 1.39(1.10) | 1.22 | 1.48(0.90) | 4.11 | 2.20(1.50) | 2.19 | 1.59(1.20) | 2.44 | 1.60(1.20) | 1.42 | 1.42(1.00) |
| 20 Newsgroups | Global Pooling CNN | 15.06 | 2.55(2.70) | 12.10 | 3.33(1.60) | 8.65 | 3.78(1.50) | 13.55 | 3.55(1.90) | 12.13 | 2.18(1.90) | 12.19 | 3.14(1.70) | 6.92 | 2.17(1.40) |
| SST Binary | Tree LSTM | 6.01 | 2.92(1.70) | 2.67 | 2.67(1.00) | 15.64 | 2.17(0.50) | 3.94 | 1.90(1.30) | 2.94 | 2.50(0.90) | 3.54 | 3.54(1.00) | 8.63 | 1.92(0.70) |

Table 8: AdaECE (%) computed for different focal loss approaches both pre and post temperature scaling (cross-validating T on AdaECE). Optimal temperature for each method is indicated in brackets.

| Dataset | Model | FL-1 | FL-2 | FL-3 | FLS-531 | FLS-532 | FLA-532 | FLA-53 |
|---|---|---|---|---|---|---|---|---|
| CIFAR-100 | ResNet 50 | 22.80 | 23.15 | 22.75 | 23.49 | 23.24 | 23.55 | 23.22 |
| | ResNet 110 | 22.36 | 22.52 | 22.92 | 22.81 | 22.96 | 22.93 | 22.51 |
| | Wide ResNet 26-10 | 19.61 | 20.01 | 19.69 | 20.13 | 20.13 | 19.71 | 20.11 |
| | DenseNet 121 | 23.82 | 23.19 | 23.25 | 23.69 | 23.72 | 22.41 | 22.67 |
| CIFAR-10 | ResNet 50 | 4.93 | 4.98 | 5.25 | 5.66 | 5.63 | 5.24 | 4.98 |
| | ResNet 110 | 4.78 | 5.06 | 5.08 | 6.13 | 5.71 | 5.19 | 5.42 |
| | Wide ResNet 26-10 | 4.27 | 4.27 | 4.13 | 4.11 | 4.46 | 4.14 | 4.01 |
| | DenseNet 121 | 5.09 | 4.84 | 5.33 | 5.46 | 5.65 | 5.46 | 5.46 |
| 20 Newsgroups | Global Pooling CNN | 26.13 | 28.23 | 29.26 | 29.16 | 28.16 | 27.26 | 27.98 |
| SST Binary | Tree LSTM | 12.63 | 12.30 | 12.19 | 12.36 | 13.07 | 12.30 | 12.80 |

Table 9: Error (%) computed for different focal loss approaches.

| Dataset | Model | Cross Entropy | Brier Loss | MMCE | FL-1 | FL-2 | FL-3 | FLS-531 | FLS-532 | FLA-532 | FLA-53 |
|---|---|---|---|---|---|---|---|---|---|---|---|
| CIFAR-100 | ResNet 50 | 153.67 | 99.63 | 125.28 | 105.61 | 92.82 | 87.52 | 100.09 | 92.66 | 94.10 | 88.03 |
| | ResNet 110 | 179.21 | 110.72 | 180.54 | 114.18 | 96.74 | 90.90 | 112.46 | 95.85 | 97.97 | 89.92 |
| | Wide ResNet 26-10 | 140.10 | 84.62 | 119.58 | 87.56 | 77.80 | 74.66 | 88.61 | 78.52 | 78.86 | 76.92 |
| | DenseNet 121 | 205.61 | 98.31 | 166.65 | 115.50 | 93.11 | 87.13 | 107.91 | 93.12 | 91.14 | 85.47 |
| CIFAR-10 | ResNet 50 | 41.21 | 18.67 | 44.83 | 22.67 | 18.60 | 18.43 | 25.32 | 20.50 | 18.69 | 17.55 |
| | ResNet 110 | 47.51 | 20.44 | 55.71 | 22.54 | 19.19 | 17.80 | 32.77 | 22.48 | 19.39 | 18.54 |
| | Wide ResNet 26-10 | 26.75 | 15.85 | 28.47 | 17.66 | 14.96 | 15.20 | 18.50 | 15.57 | 14.78 | 14.55 |
| | DenseNet 121 | 42.93 | 19.11 | 52.14 | 22.50 | 17.56 | 18.02 | 27.41 | 19.50 | 20.14 | 18.39 |
| 20 Newsgroups | Global Pooling CNN | 176.57 | 130.41 | 158.70 | 140.40 | 115.97 | 109.62 | 128.75 | 123.72 | 124.03 | 109.17 |
| SST Binary | Tree LSTM | 50.20 | 54.96 | 37.28 | 53.90 | 47.72 | 50.29 | 50.25 | 53.13 | 45.08 | 49.23 |

Table 10: NLL (%) computed for different approaches pre temperature scaling.

correctly classified samples, the model trained using focal loss has very confident predictions both pre and post temperature scaling. However, on misclassified samples, we observe a very low confidence for the focal loss model. The ResNet-50 network trained using cross entropy is very confident even

| Dataset | Model | Cross Entropy | Brier Loss | MMCE | FL-1 | FL-2 | FL-3 | FLS-531 | FLS-532 | FLA-532 | FLA-53 |
|---|---|---|---|---|---|---|---|---|---|---|---|
| CIFAR-100 | ResNet 50 | 106.83 | 99.57 | 101.92 | 94.58 | 91.80 | 87.37 | 92.77 | 91.58 | 92.83 | 88.27 |
| | ResNet 110 | 104.63 | 111.81 | 106.73 | 94.65 | 91.24 | 89.92 | 93.73 | 91.30 | 92.29 | 88.93 |
| | Wide ResNet 26-10 | 97.10 | 85.77 | 95.92 | 83.68 | 80.44 | 74.66 | 84.11 | 80.01 | 80.40 | 78.14 |
| | DenseNet 121 | 119.23 | 98.74 | 113.24 | 100.81 | 91.35 | 87.55 | 98.16 | 91.55 | 90.57 | 86.06 |
| CIFAR-10 | ResNet 50 | 20.38 | 18.36 | 21.58 | 17.56 | 17.67 | 18.34 | 19.93 | 19.25 | 17.28 | 17.37 |
| | ResNet 110 | 21.52 | 19.60 | 24.61 | 17.32 | 17.53 | 17.62 | 23.79 | 20.21 | 17.78 | 18.24 |
| | Wide ResNet 26-10 | 15.33 | 15.85 | 16.16 | 15.48 | 14.85 | 15.06 | 15.81 | 15.38 | 14.69 | 14.23 |
| | DenseNet 121 | 21.77 | 19.11 | 24.88 | 18.71 | 17.21 | 18.10 | 21.65 | 19.04 | 19.27 | 18.39 |
| 20 Newsgroups | Global Pooling CNN | 87.95 | 93.11 | 99.74 | 87.24 | 93.60 | 94.69 | 97.89 | 93.66 | 91.73 | 93.98 |
| SST Binary | Tree LSTM | 41.05 | 38.27 | 36.37 | 45.67 | 47.72 | 45.96 | 45.82 | 54.52 | 45.36 | 49.69 |

Table 11: NLL $(\%)$ computed for different approaches post temperature scaling (cross validated on ECE).

| Dataset | Model | Cross Entropy | Brier Loss | MMCE | FL-1 | FL-2 | FL-3 | FLS-531 | FLS-532 | FLA-532 | FLA-53 |
|---|---|---|---|---|---|---|---|---|---|---|---|
| CIFAR-100 | ResNet 50 | 44.34 | 36.75 | 39.53 | 33.22 | 21.03 | 13.02 | 26.76 | 23.56 | 22.40 | 16.12 |
| | ResNet 110 | 55.92 | 24.85 | 50.69 | 40.49 | 32.57 | 26.00 | 37.24 | 29.56 | 34.73 | 22.57 |
| | Wide ResNet 26-10 | 49.36 | 14.68 | 40.13 | 27.00 | 15.14 | 9.96 | 27.81 | 17.59 | 13.64 | 10.17 |
| | DenseNet 121 | 56.28 | 15.47 | 49.97 | 35.45 | 21.70 | 11.61 | 38.68 | 18.91 | 21.34 | 9.68 |
| CIFAR-10 | ResNet 50 | 38.65 | 31.54 | 60.06 | 31.75 | 25.00 | 21.83 | 30.54 | 23.57 | 25.45 | 14.89 |
| | ResNet 110 | 44.25 | 25.18 | 67.52 | 73.35 | 25.92 | 25.15 | 34.18 | 30.38 | 30.80 | 18.95 |
| | Wide ResNet 26-10 | 48.17 | 77.15 | 36.82 | 29.17 | 30.17 | 23.86 | 37.57 | 30.65 | 18.51 | 74.07 |
| | DenseNet 121 | 45.19 | 19.39 | 43.92 | 38.03 | 29.59 | 77.08 | 33.50 | 16.47 | 17.85 | 13.36 |
| 20 Newsgroups | Global Pooling CNN | 36.91 | 31.35 | 34.72 | 34.28 | 24.10 | 18.85 | 26.02 | 25.02 | 24.29 | 17.44 |
| SST Binary | Tree LSTM | 71.08 | 92.62 | 68.43 | 95.48 | 86.21 | 22.32 | 76.28 | 86.93 | 80.85 | 73.70 |

Table 12: MCE $(\%)$ computed for different approaches pre temperature scaling.

| Dataset | Model | Cross Entropy | Brier Loss | MMCE | FL-1 | FL-2 | FL-3 | FLS-531 | FLS-532 | FLA-532 | FLA-53 |
|---|---|---|---|---|---|---|---|---|---|---|---|
| CIFAR-100 | ResNet 50 | 12.75 | 21.61 | 11.99 | 8.92 | 8.86 | 6.76 | 7.46 | 6.76 | 5.24 | 27.18 |
| | ResNet 110 | 22.65 | 13.56 | 19.23 | 20.13 | 12.00 | 13.06 | 18.28 | 13.72 | 15.89 | 10.94 |
| | Wide ResNet 26-10 | 14.18 | 13.42 | 16.50 | 10.28 | 18.32 | 9.96 | 13.18 | 11.01 | 12.50 | 9.73 |
| | DenseNet 121 | 21.63 | 8.55 | 13.02 | 10.49 | 11.63 | 6.17 | 6.21 | 6.48 | 9.41 | 5.68 |
| CIFAR-10 | ResNet 50 | 20.60 | 22.46 | 23.60 | 25.86 | 28.17 | 15.76 | 22.05 | 23.85 | 24.76 | 26.37 |
| | ResNet 110 | 29.98 | 22.73 | 31.87 | 29.74 | 23.82 | 37.61 | 26.25 | 25.94 | 11.59 | 17.35 |
| | Wide ResNet 26-10 | 26.63 | 77.15 | 32.33 | 74.58 | 29.58 | 25.64 | 28.63 | 20.23 | 19.68 | 36.56 |
| | DenseNet 121 | 32.52 | 19.39 | 27.03 | 19.68 | 22.71 | 76.27 | 21.05 | 32.76 | 35.06 | 13.36 |
| 20 Newsgroups | Global Pooling CNN | 6.10 | 13.04 | 15.12 | 7.79 | 11.88 | 22.22 | 16.86 | 11.85 | 8.37 | 12.65 |
| SST Binary | Tree LSTM | 88.48 | 91.86 | 32.92 | 87.77 | 86.21 | 74.52 | 54.27 | 88.85 | 82.42 | 76.71 |

Table 13: MCE $(\%)$ computed for different approaches post temperature scaling (cross validated on ECE).

on the misclassified samples, particularly before temperature scaling. Apart from focal loss, the only model which has relatively low confidences on misclassified test samples is the one trained using Brier score. These observations support our claim that focal loss produces not only a calibrated model but also one which is confident on its correct predictions.

## G  FOCAL LOSS AND CROSS ENTROPY ON A LINEAR MODEL

To understand the effect of focal loss on simpler to analyse models, we conducted experiments on a generalised linear model on a simple data distribution.

**Setup**  We consider a binary classification problem. The data matrix $\mathbf{X} \in \mathbb{R}^{2 \times N}$ is created by assigning each class two normally distributed clusters such that the mean of the clusters are linearly separable. The mean of the clusters are situated on the vertices of a two-dimensional hypercube. Further to that, for $10\%$ of the data points, the labels were flipped. The model consists of a simple 2-parameter logistic regression model. $f_{\mathbf{w}}(\mathbf{x}) = \sigma(w_1 x_1 + w_2 x_2)$. We train this model using both cross entropy and focal loss with $\gamma = 1$.

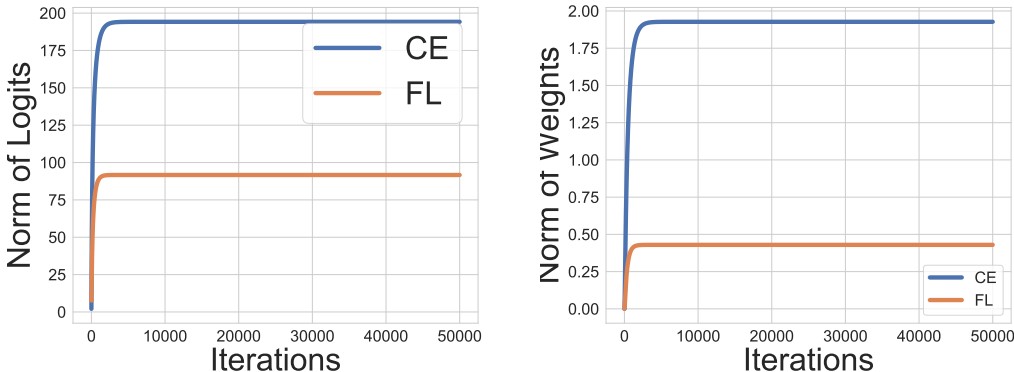

Figure 5: (a): Norm of logits (b): Norm of weights.

**Weight Magnification** We have argued that focal loss implicitly regularizes the weights of the model by providing smaller gradients as compared to cross entropy. This helps in calibration as - if all the weights are large, the logits are large and thus the confidence of the network is large on all test points. When the model misclassifies, it misclassifies with a high confidence. Figure 5 shows, for a generalised linear model, that the norm of the logits and the weights of a network blows for Cross Entropy as compared to Focal Loss.

**High Confidence for mistakes** Figure 6 (b) and (c) shows that running gradient descent with cross entropy (CE) and focal loss (FL) both gives the same decision regions i.e. the weight vector points in the same region for both FL and CE. However, as we have seen that the norm of the weights is much larger for FL as compared to CE, we would expect the confidence of misclassified test points to be large for CE as compared to FL. Figure 6 (a) plots a histogram of the confidence of the misclassified points and it shows that CE misclassifies almost always with greater than 90% confidence whereas FL misclassifies with much lower confidence.

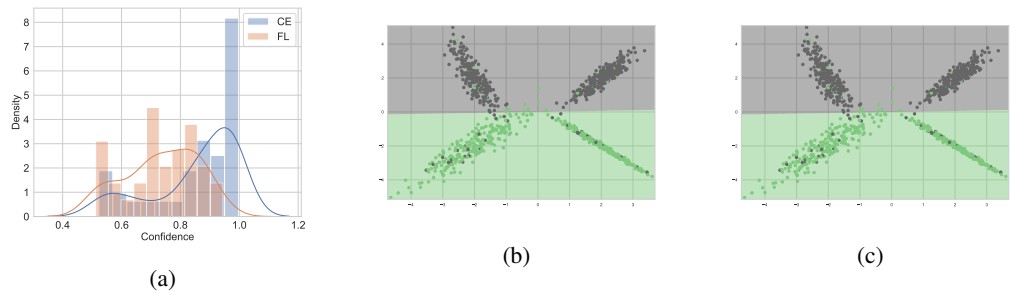

Figure 6: (a): Confidence of mis-classifications (b): Decision boundary of linear classifier trained using cross entropy (c): Decision boundary of linear classifier trained using focal loss

# H    FOCAL LOSS MINIMISES REGULARISED BREGMAN DIVERGENCE

In this section, we show that just like Cross Entropy minimizes an upper bound on the KL-Divergence between the true distribution $p$ and the predicted distribution $q$, focal loss minimizes a regularized KL divergence between $p$ and $q$. We use $FL_\gamma(p, q)$ and $CE(p, q)$ to denote the focal loss with parameter

$\gamma$ and cross entropy between $p$ and $q$ respectively and $K$ to denote the number of classes.

$$FL_\gamma(p,q) = -\sum_i^K (1-q_i)^\gamma p_i \log q_i$$

$$\geq -\sum_i^K (1-\gamma q_i) p_i \log q_i \qquad \text{By Bernoulli's inequality } \forall \gamma \geq 1 \text{ and } 0 \leq q_i \leq 1$$

$$= -\sum_{i=1}^K p_i \log q_i - \gamma \left| \sum_{i=1}^K p_i q_i \log q_i \right| \qquad \forall i \log q_i \leq 0$$

$$\geq -\sum_{i=1}^K p_i \log q_i - \gamma \max_j p_j \sum_{i=1}^K |q_i \log q_i| \qquad \text{By Holder's inequality with } p = \infty, q = 1$$

$$\geq -\sum_{i=1}^K p_i \log q_i + \gamma \sum_{i=1}^K q_i \log q_i \qquad \forall i \ p_i \leq 1$$

$$= CE(p,q) - \gamma \text{Entropy}(q)$$

We know that

$$\text{KL}(p||q) = CE(p,q) - \text{Entropy}(p)$$
$$\text{KL}(p||q) \leq \gamma \text{Entropy}(q) + FL(p,q) - \text{Entropy}(p)$$
$$FL(p,q) \geq \text{KL}(p||q) + \text{Entropy}(p) - \gamma \text{Entropy}(q)$$

Experimentally, we found the solution of the Cross Entropy and Focal loss equation, i.e. the value of the predicted probability $\hat{q}$ which minimizes the loss, for various values of $p$ in a binary classification problem (i.e. $K = 2$) and plotted it in Figure 7. Note how Focal loss favours a more entropic solution $\hat{q}$ that is closer to 0.5. In other words, as Figure 7 shows, solutions to focal loss (Eqn equation 13) will always have higher entropy than that of Cross Entropy.

$$\hat{q} = \text{argmin}_x - (1-x)^\gamma p \log x - x^\gamma (1-p) \log(1-x) \quad 0 \leq x \leq 1 \tag{13}$$

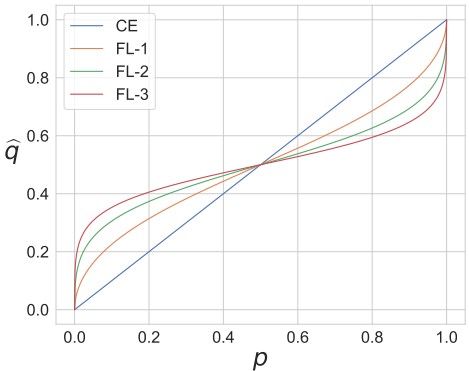

Figure 7: Optimal $\hat{q}$ for various values of $p$ and $\gamma$

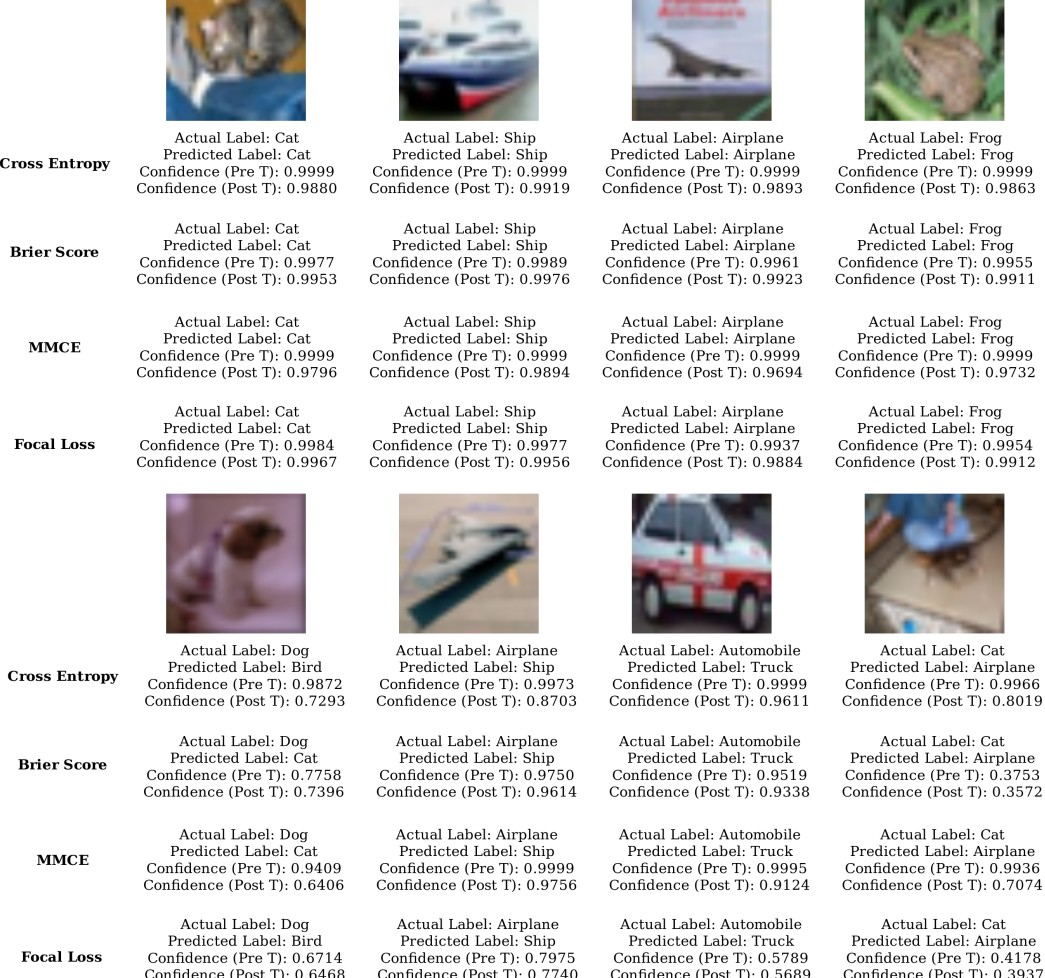

Figure 8: Qualitative results showing the performance of Cross Entropy, Brier Score, MMCE and Focal Loss (sample-wise $\gamma$ 5,3) for a ResNet-50 trained on CIFAR-10. The first row of images have been correctly classified by networks trained on all four loss functions and the second row of images have all been incorrectly classified. For each image, we present the actual label, the predicted label and the confidence of the prediction both before and after temperature scaling.

