# OpenReview forum: "The Intriguing Effects of Focal Loss on the Calibration of Deep Neural Networks"
_ICLR.cc/2020/Conference — Reject_

### Official Review · AnonReviewer1 · 2019-10-10
**Official Blind Review #1**

**Rating:** 3

**Review:**

The paper explores how focal loss can be used to improve calibration for classifiers. Focal loss extends the cross-entropy loss, which is -log(p_label), with a multiplicative factor equal to (1 - p_label)^gamma. Intuitively, this downweights the loss for elements where the probability of the correct label p_label is close to 1, relatively increasing the weight of the misclassified examples.

Somewhat surprisingly, this tends to improve the calibration of the model. I say surprisingly because the focal loss is not a bregman divergence for all values of alpha so in general the expected minimizer of the focal loss for a fractional label is not the fractional label (i.e. the minimizer wrt x of - p (1-x)^gamma log(x) - (1-p) x^gamma log (1 -x) is not in general p).

The paper shows somewhat thorough experiments on many datasets justifying this observation, but the theoretical part is rather weak since it doesn't seem to address this issue with the focal loss.

It's also not very clear from reading the paper what the p0 should be when using the rule to automatically select the gamma of the focal loss.

I'd support accepting the paper if the calibration properties of the focal loss itself was better analyzed on a simpler setup (linear models, or single parameter models) so it's easier to understand how it's helping calibration in the deep network setup and if the algorithm for choosing per-example gammas was more clearly stated out.

**Experience Assessment:**

I have published one or two papers in this area.

**Review Assessment: Checking Correctness Of Derivations And Theory:**

I carefully checked the derivations and theory.

**Review Assessment: Checking Correctness Of Experiments:**

I assessed the sensibility of the experiments.

**Review Assessment: Thoroughness In Paper Reading:**

I read the paper thoroughly.

---

> ### Author Response · Authors · 2019-11-14
> **Response to R1 Comment 1: Focal loss not a Bregman Divergence**
>
> We sincerely thank the reviewer for this comment which we address as follows.
>
> We deal with a classification problem where $p=0$ or $p=1$ (with one-hot encodings) and expected minimizer of focal loss for it comes out to be $x=p$.  However, we believe that the following question needs to be addressed:
>
> Q. Focal loss is not a bregman divergence, thus the minimizer of focal loss is not the original label when the original label is fractional. So, what exactly is it minimizing ?
>
> Ans: Yes focal loss is not a bregman divergence. However, in Appendix H we show that it is a regularized bregman divergence in the sense that  while $\mathrm{CrossEntropy}(p,q)
>  = \mathrm{KL}(p||q) + \mathrm{Entropy}(p)$, we have $\mathrm{FocalLoss}(p,q) > \mathrm{KL}(p||q) + \mathrm{Entropy}(p) - \gamma * \mathrm{Entropy}(q)$. Thus it is minimizing the KL-divergence between the target and predicted label distribution while ensuring that the entropy of the predicted distribution is large and $\gamma$ is essentially the regularization coefficient. Having higher entropy on the predicted distribution can help avoid overconfident predictions observed in modern neural networks, thus leading to better calibration. We provide the related proof in Appendix H.
>
> Please let us know if this answers your question or if not, it would be very helpful if you could give us some more clarity about what you are looking for in the theoretical part.

---

> > ### Comment · AnonReviewer1 · 2019-11-15
> > **response**
> >
> > (there are way too many author responses here, so trying to consolidate my response-to-the-response)
> >
> > First, sorry, I used the term bregman divergence in my review when I meant to refer to proper loss (which is a loss where the minimizer is the true probability).
> >
> > The point about focal loss being cross entropy + regularization on the entropy of the predictive distribution is interesting, thanks for including it.
> >
> > It still feels weird to me that the paper claims that focal loss improves calibration when in fact focal loss + perfect optimization in the scenario where we have label noise will in fact calibrate predictions incorrectly (and this is true regardless of whether the model is linear or deep). It seems clear to me that there is something good happening here because of the focal loss, as the experimental results do show improved calibration, but I still don't feel like I understand why.
> >
> > In a way what seems to happen is that models are likely to end up in a regime where the entropy of the predictive distribution is lower than the entropy of the actual target distribution, and so regularizing this entropy by some amount will counter this tendency and lead to better-calibrated models. But why do models tend to have sharper-than-real predictive distributions? Is this overfitting? Is this an artifact of SGD optimization? Understanding why this happens would be helpful in understanding why this regularization is needed.
> >
> > Still, despite the explanations in the paper and the comment above, I still don't see how you can go from "we're regularizing the predictive distribution's entropy" to "this is the optimal regularization value for this sample"; the assumptions behind that optimization used to pick the best gamma are not clearly stated. Please clarify.

---

> > > ### Author Response · Authors · 2019-11-15
> > > **Response to R1's response (Part 2)**
> > >
> > > >> Still, despite the explanations in the paper and the comment above, I still don't see how you can go from "we're regularizing the predictive distribution's entropy" to "this is the optimal regularization value for this sample"; the assumptions behind that optimization used to pick the best gamma are not clearly stated. Please clarify.
> > >
> > > --We are really sorry for the confusion but we don’t claim in the paper or in our responses that the value of $\gamma$ we use for the focal loss sample-wise $\gamma$ approach (or for any of the focal loss approaches like constant $\gamma$ or scheduled $\gamma$) is “the optimal regularization value”. We design the policies for sample-wise $\gamma$ using certain observations (we have listed these in points 1,2 and 3 in our response to your comment 4) and we clarify our rationale behind the policies in our response to your comment 4.
> > >
> > > Empirically, we find the designed policies to perform very well across all datasets and models we trained on and hence, we propose these policies in the paper. Having said that, these policies are heuristics and we cannot claim that the values of $\gamma$ we use are optimal. In fact, as we mentioned in our responses, we are actually interested in finding a more principled approach/algorithm to design these policies for future work.
> > >
> > > Finally, focal loss is an upper bound on KL divergence - entropy * $\gamma$ (not equality), so $\gamma$ is the regularization coefficient of the entropy of the predicted distribution. On the other hand, we design our strategy for choosing sample-wise $\gamma$ using Propositions 1 and 2 to minimize gradient norm. These are two different interpretations of $\gamma$ and we don’t try to make any leap from one to the other.

---

> > > ### Author Response · Authors · 2019-11-15
> > > **Response to R1's response (Part 1)**
> > >
> > > Thank you very much for your quick response, below we provide our replies.
> > >
> > > >> First, sorry, I used the term bregman divergence in my review when I meant to refer to proper loss (which is a loss where the minimizer is the true probability).
> > >
> > > -- Thank you for the clarification. We agree that focal loss is not a proper loss function which can also be inferred from the Figure 7 in the appendix. However, as we have mentioned in our response to your comment 1, in our classification experimental setup there is no label noise, and we train on one-hot encodings. Hence, the expected minimizer of focal loss comes out to be $x = p$. We will make it clear in our final submission.
> > >
> > > >>The point about focal loss being cross entropy + regularization on the entropy of the predictive distribution is interesting, thanks for including it.
> > >
> > > -- Thank you.
> > >
> > >
> > > >> It still feels weird to me that the paper claims that focal loss improves calibration when in fact focal loss + perfect optimization in the scenario where we have label noise will in fact calibrate predictions incorrectly (and this is true regardless of whether the model is linear or deep). It seems clear to me that there is something good happening here because of the focal loss, as the experimental results do show improved calibration, but I still don't feel like I understand why.
> > >
> > > -- Thanks for pointing this out. We agree that if we have label noise, perfect optimization leads to optimal fractional predicted probability depending on the level of noise even in the case of a classification problem and because focal loss is not a proper loss, it won’t be calibrated in case of noisy labels. However, there is no label noise in our classification experimental setup. Also, standard optimization methods which are normally used in modern deep neural networks do not lead to perfect optimization so empirically it may be tough to compare focal loss with cross entropy in case of noisy labels. We will make it clear in the main paper. In our experimental setup we consider standard classification datasets without any noisy labels. This, along with other justifications made in the paper indicate why focal loss performs better.
> > >
> > >
> > > >> In a way what seems to happen is that models are likely to end up in a regime where the entropy of the predictive distribution is lower than the entropy of the actual target distribution, and so regularizing this entropy by some amount will counter this tendency and lead to better-calibrated models. But why do models tend to have sharper-than-real predictive distributions? Is this overfitting? Is this an artifact of SGD optimization? Understanding why this happens would be helpful in understanding why this regularization is needed.
> > >
> > > Training a model on separable data with loss functions like NLL (which contains an exponential map) will give a minimizer where the norms of its parameters are infinity (For a proof with linearly separable data one could look at Lemma 1 in [Soudry et. al.]. In neural networks, one can consider the learnt representations before the last linear layer to be linearly separable as neural networks often achieve near perfect training accuracy.)  Thus, by optimizing this loss, gradient descent increases the parameter norms.
> > >
> > > When the parameters blow up, the logits will blow up and thus the predictive distribution will be very sharp. For focal loss, this happens much slower than it happens for cross entropy (or NLL) as we show that the gradient norms are smaller and thus the weights blow up slower. We argue that this combination of exponential loss (minimizer at infinity) and high-capacity model (hence separable data) causes this low entropy of the predictive distribution. On a similar note, it was also observed in Guo et. al.  that neural networks can overfit to NLL without overfitting to the 0/1 loss (refer to Entropy before calibration in Fig S1 in Supplementary in Guo. et. al.).
> > >
> > >
> > > Soudry, Daniel, et al. "The implicit bias of gradient descent on separable data." The Journal of Machine Learning Research 19.1 (2018): 2822-2878.
> > > Guo, Chuan, et al. "On calibration of modern neural networks." Proceedings of the 34th International Conference on Machine Learning-Volume 70. JMLR. org, 2017.

---

> ### Author Response · Authors · 2019-11-14
> **Response to R1 Comment 2: Value of p0 when using the rule to automatically select gamma for focal loss**
>
> We sincerely thank the reviewer for this comment which we address as follows.
>
> We use $p_0$ as a notation to indicate the probability for which we find the $\gamma$ such that $g(p_0, \gamma) = 1$ (refer Proposition 2). Hence, in the context of automatically selecting $\gamma$ for focal loss, when we say $p_0$, we are assuming that the reviewer means the probability values at which we change $\gamma$ in the policies for the sample-wise $\gamma$ approach. For instance, in the Focal Loss (sample-wise $\gamma$ 5,3) policy, we have $p_0 = 0.19$ where we change $\gamma$ from 5 to 3 and for the Focal Loss (sample-wise $\gamma$ 5,3,2) policy we have two $p_0$s, one at 0.19 where we change $\gamma$ from 5 to 3 and the other one at 0.5 where we change $\gamma$ from 3 to 2. In our experiments, we use the following general rules for selecting $p_0$ values for the policies:
>
> 1. One of the changing points to consider should be $p = 0.5$. For any $p > 0.5$, we want $g(p, \gamma) < 1$.
>
> 2. In the interval $[0, 0.5]$, we do not want $g(p, \gamma)$ to be much lower than 1. In fact, for lower probability values in the interval $[0, 0.5]$, we want $g(p, \gamma)$ to be higher than 1. Therefore, we divide the interval $[0, 0.5]$ into sub-intervals of the form $[a, b]$ and choose a $\gamma$ for that sub-interval such that $g(b, \gamma) = 1$. Hence, for all probability values $p$ lying in the interval $[a, b]$, $g(p, \gamma) > 1$.
>
> We designed a few policies based on the above two rules and chose the ones which consistently performed the best on validation sets. However, as part of our future work in this direction, we’re interested in designing an algorithm which can provide an optimal policy for any dataset, network pair.

---

> ### Author Response · Authors · 2019-11-14
> **Response to R1 Comment 3: Analysing calibration properties of focal loss on a simpler setup**
>
> We sincerely thank the reviewer for this comment which we address as follows.
>
> The behaviour of deep neural networks is generally quite different from linear models and the problem of calibration is more pronounced in the case of deep neural networks. Hence, we focus on analysing the calibration of deep networks in the paper. We have argued in the paper that the increase in magnitude of weights in the network during training is one of the main reasons for miscalibration. This increase in magnitude of weights leads to an increase in the norm of the logits of the network on all points irrespective of whether the point is correctly or incorrectly classified. In case of focal loss, the increase in the magnitude of weights is much lower as compared to cross entropy and hence, models trained using focal loss are more calibrated.
>
> We analyse this setup using a linear model trained on linearly separable data with some noise using both cross entropy and focal loss in Appendix G. We observe that the norm of the logits and in turn, the magnitude of weights increases during training on a linear model, which is not very surprising as exponential losses like cross entropy on linearly separable data have their only minimizer at infinity. However, as our experiments show, this magnification of norms (of weight and logits) is significantly higher for the cross entropy loss as compared to the focal loss though both the losses produce the same decision boundary (shown in Figures 5, 6(b) and 6(c) in the Appendix). Also consistent with our argument, this leads to higher confidence for misclassified points in the case of cross entropy as compared to focal loss (Figure 6(a) in the appendix). In short, the linear model shows that, for our data (which is separable for the linear model), training with cross entropy loss leads to higher logit norms and weight norms thereby producing higher confidence for misclassified points as compared to focal loss.

---

> ### Author Response · Authors · 2019-11-14
> **Response to R1 Comment 4: Choosing per-example gammas**
>
> We sincerely thank the reviewer for this comment which we address as follows.
>
> We design the policies for sample-dependent $\gamma$ based on three observations:
> 1. Using Proposition 2 in the paper, given a prediction confidence $p_0$, we can compute a value of $\gamma$ (say $\gamma^*$) such that $g(p_0, \gamma^*) = 1$.
>
> 2. From Proposition 2 (and visually from Figure 4(a)), we know that for the same $\gamma^*$, if we choose a $p$ such that $p < p_0$, we get $g(p, \gamma^*) > 1$ and if we choose a $p > p_0$, we get $g(p, \gamma^*) < 1$.
>
> 3. Finally, from Proposition 1, we know that if $g(p, \gamma) < 1$, the gradient norm of focal loss is lower than that of cross entropy and if $g(p, \gamma) > 1$, the gradient norm of focal loss is higher than that of cross entropy. Hence, the weight regularisation effect of focal loss is present only when $g(p, \gamma) < 1$.
>
>
> For a given training sample, $(x_i, y_i)$, let us say that $p_i$ is the confidence of the model on the correct class. If $p_i > 0.5$, we can say for certain that the sample has been correctly predicted. Hence, we want to accelerate the rise in the network’s confidence as long as $p_i < 0.5$. However, at the same time, we also want to regularise the weights of the network so that they don’t blow up leading to the network becoming overconfident and consequently, miscalibrated.
>
>
> Keeping these requirements in mind, we design the policy for $\gamma$ such that when $p_i \in [0, 0.25]$, $g(p_i, \gamma) > 1$ and when $p_i \in (0.25, 1]$, $g(p_i, \gamma) < 1$. This ensures that when the network has a very low confidence on the correct class (confidence being lower than 0.25), we are making the network more confident by ensuring that the gradient norms are higher (as compared to cross entropy) as we have $g(p_i, \gamma) > 1$ (see Proposition 1). However, when the network reaches a confidence of 0.25 or more on the sample, we start regularising the weights of the network by having $g(p_i, \gamma) < 1$.
>
>
> It should be noted that in the interval $p_i \in [0, 0.25]$, we want $g(p_i, \gamma)$ to be observably higher than 1 (i.e., we don’t want $g(p_i, \gamma) \approx 1$). As can be seen from Figure 4(a), a higher value of $\gamma$ in this interval provides a higher value of $g(p_i, \gamma)$.  Hence, we set $\gamma$ to 5 for low confidence values $p_i \in [0, 0.19]$ as $g(p_i, 5) \approx 1$ at $p_i = 0.19$. For $p_i > 0.19$, we change the $\gamma$ from 5 to 3 so that $g(p_i, \gamma)$ stays above 1 for $p_i < 0.25$ (as $g(0.25, 3) \approx 1$, also refer Figure 4(a)).
>
>
> Finally, when $p_i \in (0.25, 1]$, we don’t want to set a high value of $\gamma$ as such a value can lead to a steep drop in $g(p_i, \gamma)$ thereby causing the gradients to vanish. Thus, we either stick with 3 (Focal Loss (sample-wise $\gamma$ 5,3) policy) or change $\gamma$ to 2 for $p_i > 0.5$ (Focal Loss (sample-wise $\gamma$ 5,3,2) policy).
>
>
> We found the above mentioned policies (especially the Focal Loss (sample-wise $\gamma$ 5,3) policy) to consistently perform the best across all the network architectures we tried and across all the datasets we trained on. In fact, it also performed really well on our Tiny ImageNet experiment (results for which can be found in our response to R2’s comment 1). Having said that, we think that an optimal policy which can be easily computed for any model and dataset combination would be ideal and would remove the need to hand-design policies. Developing an algorithm to compute such an optimal policy for any dataset-model pair is something we are interested in pursuing as future work.
>
>
> We will clarify the above mentioned points in the main paper.

---

### Official Review · AnonReviewer2 · 2019-10-18
**Official Blind Review #2**

**Rating:** 6

**Review:**

Summary:
This paper studies the effect of the focal loss, proposed by Lin et al. in 2017 on network miscalibration, which appears when the network's confidence in its prediction does not match its correctness. The authors provide a theoretical explanation to the superior results of the focal loss for calibration. The temperature scaling technique of Guo et al. 2017 is applied (dividing the network's logits by a scalar learnt on a val set prior to softmax) to networks trained using the focal loss, with different options for the focal parameter, as well as the standard multi-class cross entropy and a few others. The experiments on CIFAR10/100 as well as two text dataset (20 Newsgroups, Stanford Sentiment Treebank) reach lower expected calibration error compared to the cross entropy (75% of relative improvement on cifar100 for instance).

The importance of the contribution will probably be discussed here. At first glance, it seems that the works build mainly on advances from Lin et al & Guo et al, but the authors do a promising job in combining the two.

Positive aspects:
- The paper is well written.
- Experiments on both image and text dataset demonstrate the superiority of the focal loss on several calibration metrics.
- The theoretical explanation is convincing.

Negative points:
- The importance of the problem is motivated by future assessments by downstream tasks but do not address this aspect in the experiments. In particular, as the images experiments are conducted on tiny images, an experiment on a real size image dataset would strengthen the paper.
- The policy that works best for defining the sample wise tuning of the focal parameter was hand-made but ultimately uses only 3 parameters so finally it is not so bad.

Minor:
- It'd be nice to illustrate the confidence improvements on a few qualitative examples, maybe in appendix.
- 10 pages is too much (given that were were given instructions to be more severe with long paper) table 6 and 3 could be merged for instance.
- The focal loss column results of table 1 should be the same as Table 5 (sample wise)?
- could specify what MMCE means
- clean the bibliography

I've read the other reviews and authors' responses. Experiences on Tiny ImageNet are better than CIFAR but still a little far from what I'd call real images but I understand it can be difficult to run experiments on ImageNet. Since the choice of gamma seems to be leading consistent results also on tinyIN, I find it less concerning.

**Experience Assessment:**

I have read many papers in this area.

**Review Assessment: Checking Correctness Of Derivations And Theory:**

I assessed the sensibility of the derivations and theory.

**Review Assessment: Checking Correctness Of Experiments:**

I carefully checked the experiments.

**Review Assessment: Thoroughness In Paper Reading:**

I read the paper thoroughly.

---

> ### Author Response · Authors · 2019-11-14
> **Response to R2 comment 1: Experiments done on tiny images**
>
> We sincerely thank the reviewer for this comment which we address as follows:
>
> To compare the performance on a bigger image dataset, we trained the ResNet-50 network using cross entropy, focal loss with a fixed gamma value of 3 and focal loss with the sample-wise gamma policy of 5,3 on Tiny ImageNet. The Tiny ImageNet dataset is a subset of ImageNet with 64 x 64 dimensional images, 200 classes and 500 images per class in the training set and 50 images per class in the validation set. The image dimensions of Tiny ImageNet are twice the size of the CIFAR-10/100 dataset images.
>
> We use SGD with a momentum of 0.9 as our optimiser and train the networks for 100 epochs with a learning rate of 0.1 for the first 40 epochs, 0.01 for the next 20 epochs and 0.001 for the last 40 epochs. We use a training batch size of 64. We also augment the training images with random crops and random horizontal flips. It should be noted that we saved 50 samples per class (i.e., a total of 10000 samples) from the training set as our own validation set to fine-tune the temperature parameter on (hence, we trained on 90000 images) and we use the Tiny ImageNet validation set as our test set. We report the Tiny ImageNet validation set error%, ECE% both before and after temperature scaling and Ada-ECE% both before and after temperature scaling in the table below.
>
> +------------------------------------+----------+----------------+-----------------+--------------------+---------------------+
> |           Loss Function            | Error(%) | ECE(%) (Pre T) | ECE(%) (Post T) | Ada-ECE(%) (Pre T) | Ada-ECE(%) (Post T) |
> +------------------------------------+----------+----------------+-----------------+--------------------+---------------------+
> | Cross Entropy                                          |    49.88 |          14.98 | 5.05(1.4)       |              14.98 | 5.05(1.4)           |
> | Focal Loss (gamma = 3)                         |    48.37 |           2.08  | 2.08(1.0)       |               1.71  | 1.71(1.0)           |
> | Focal Loss (Sample-wise gamma-5,3) |    48.43 |           1.80  | 1.80(1.0)       |               2.06  | 2.06(1.0)           |
> +------------------------------------+----------+----------------+-----------------+--------------------+---------------------+
>
> Firstly, we observe that models trained on focal loss have a lower error than the model trained using cross entropy (thus better accuracy).
>
> Secondly, we also observe a significant improvement in ECE and Ada-ECE values both before and after temperature scaling for models trained on focal loss indicating that these models are not only more accurate but also much more calibrated.
>
> Finally, we note that the optimal temperature for both models trained using focal loss is 1 which indicates that temperature scaling could not make these models any more calibrated.
>
> We will add all these baselines to the paper. We report these preliminary results here (and not in the paper) because we have to train quite a few other networks (like ResNet-110, DenseNet, Wide ResNet etc.) and also we have to train other baselines (like MMCE, Brier Score and other versions of focal loss) to obtain the complete set of results which we can then add to the paper.

---

> ### Author Response · Authors · 2019-11-14
> **Response to R2 Minor Comments**
>
> We sincerely thank the reviewer for these comments which we address as follows:
>
> 1. "It'd be nice to illustrate the confidence improvements on a few qualitative examples, maybe in appendix": We have added some qualitative results to show the confidence improvements of focal loss in the Appendix F (Figure 8). We took ResNet-50 networks trained on CIFAR-10 using cross entropy, MMCE, Brier score and focal loss (with sample-wise gamma 5,3) and presented the prediction confidence of these networks on correctly and incorrectly classified test samples. We have reported the confidence estimates obtained both before and after temperature scaling. The observations we make in Appendix F support the claim that models trained using focal loss are well calibrated and confident on their correct predictions.
>
> 2. "10 pages is too much (given that were were given instructions to be more severe with long paper) table 6 and 3 could be merged for instance": We have merged Tables 3 and 6 in the paper now.
>
> 3. "The focal loss column results of table 1 should be the same as Table 5 (sample wise)?":
> Briefly, Table 1 compares the loss functions over ECE% while Table 5 compares them over Adaptive ECE%, hence the results are not necessarily the same.
>
> In Table 1, we report ECE(%) values of all the baselines (Cross Entropy, Brier Loss, MMCE) along with the best approach we found among all the focal loss approaches (i.e., sample-wise focal loss). In Table 2, we report the Adaptive ECE(%) values for the same frameworks. Tables 4 and 5, on the other hand, are meant to compare the different focal loss approaches over ECE and Adaptive ECE respectively. Hence, the focal loss column results of Table 1 are the same as the Focal loss sample-wise column in Table 4. Similarly, the focal loss column results in Table 2 are the same as the Focal Loss sample-wise column in Table 5. Since the purposes of Tables 1, 2 and Tables 4, 5 are different, we have kept both sets of tables in the paper.
>
> 4. "could specify what MMCE means": We have mentioned the meaning of MMCE in Section 1 (Introduction) of the paper (it can be found in the second paragraph of Page 2). Furthermore, in the updated version of the paper, we have included a two-sentence description of MMCE in Section 5 (Experiments) in the Baselines (Cross Entropy, MMCE and Brier Score) paragraph.
>
> 5. "clean the bibliography": We have updated the references, which previously had arXiv links for some papers that have actually been accepted and published. The updated references now provide information about the conferences/journals in which the papers were published. Please let us know if there are any other changes you’d like us to make to the bibliography.

---

> ### Author Response · Authors · 2019-11-14
> **Response to R2 comment 2: Policies to choose gamma**
>
> We sincerely thank the reviewer for this comment which we address as follows.
>
> In response to R1’s comment 2 and comment 4, we have mentioned the intuitive ways in which we designed the simple policies for sample-wise gamma. We agree that the policies were hand-made but they were theoretically motivated using certain observations which we state in our response (to R1). As part of our future work, we want to develop a more principled algorithm to design these policies.

---

### Official Review · AnonReviewer3 · 2019-10-20
**Official Blind Review #3**

**Rating:** 6

**Review:**

The paper describes how the use of the now-standard focal loss can lead to improved calibration results when used to fit deep-models. When fitting a large capacity model with NLL, the model can often try to drive its predictions close to 1 (i.e. infinity pre-softmax) on the training set, ultimately leading to poorly calibrated models and overfitting behaviour. The focal loss appears to mitigate this issue.

The approach is extremely simple to implement, the theoretical justifications are believable, and the calibration/accuracy performances seem to be good -- for this reasons, I think that the paper should be accepted.

(1) it would be interesting to compare the approach to using the standard cross-entropy applied to smoothed labels (i.e. (1-eps,eps) instead of (1,0) in binary classification and obvious generalisation in multi-class setting).

(2) data-augmentation often greatly helps with calibration -- the paper did not describe in details what has been done on that front for the numerical investigations.

**Experience Assessment:**

I have read many papers in this area.

**Review Assessment: Checking Correctness Of Derivations And Theory:**

I carefully checked the derivations and theory.

**Review Assessment: Checking Correctness Of Experiments:**

I carefully checked the experiments.

**Review Assessment: Thoroughness In Paper Reading:**

I read the paper at least twice and used my best judgement in assessing the paper.

---

> ### Author Response · Authors · 2019-11-14
> **Response to R3 Comment 1: Comparison with cross entropy applied to smoothed labels (Part 1)**
>
> We sincerely thank the reviewer for this comment which we address as follows.
>
> To empirically observe the effects of training networks using cross entropy loss with smoothed labels, we trained ResNet-50 and ResNet-110 on both CIFAR-10 and CIFAR-100 using cross entropy loss with smoothing factors of 0.05 and 0.1. In simple terms, if the smoothing factor is $\alpha$ and if for a sample we have a one-hot label vector $Y$, then its smoothed label vector $S$ will be such that $S_i = (1 - \alpha ) * Y_i + \alpha * (1 - Y_i) / (K-1)$ where $K$ is the number of classes. In Table 1, we present the test set error %, ECE (%) both pre and post temperature scaling and Ada-ECE(%) both pre and post temperature scaling for each of these configurations.
>
> We also provide the same metrics for ResNet-50 and ResNet-110 trained using cross entropy and focal loss with one-hot labels (these numbers were taken from Tables 1,2 and 3 in the paper) for ease of comparison in Table 2. We provide Table 2 in a separate comment due to lack of space. The focal loss numbers in Table 2 are for focal loss with the sample-wise gamma approach reported in Tables 1 and 2 of the paper.
>
> Table 1
> +-------+-----------+-----------+------------+----------+----------------+-----------------+--------------------+---------------------+
> | Loss  | Smoothing |  Dataset  |   Model    | Error(%) | ECE(%) (Pre T) | ECE(%) (Post T) | Ada-ECE(%) (Pre T) | Ada-ECE(%) (Post T) |
> +-------+-----------+-----------+------------+----------+----------------+-----------------+--------------------+---------------------+
> | CE    |      0.05 | CIFAR-10   | ResNet-50   |     4.99  |           3.08  | 1.33(0.9)       |               3.74  | 2.89(0.8)           |
> | CE    |      0.05 | CIFAR-10   | ResNet-110 |     5.11  |           1.56  | 1.82(0.9)       |               3.23  | 2.52(0.9)           |
> | CE    |      0.05 | CIFAR-100 | ResNet-50   |    22.10 |           7.61  | 4.19(1.1)       |               7.80  | 6.32(1.1)           |
> | CE    |      0.05 | CIFAR-100 | ResNet-110 |    23.45 |          10.89 | 5.97(1.1)       |              10.71 | 7.77(1.1)           |
> | CE    |       0.1  | CIFAR-10   | ResNet-50   |     5.04  |           7.30  | 1.02(0.8)       |               7.90  | 2.95(0.8)           |
> | CE    |       0.1  | CIFAR-10   | ResNet-110 |     5.26  |           6.55  | 1.27(0.8)       |               8.09  | 3.20(0.8)           |
> | CE    |       0.1  | CIFAR-100 | ResNet-50   |    22.82 |           5.27  | 5.27(1.0)       |               5.93  | 5.93(1.0)           |
> | CE    |       0.1  | CIFAR-100 | ResNet-110 |    22.51 |           4.55  | 4.55(1.0)       |               8.29  | 8.29(1.0)           |
> +--------+-----------+-----------+------------+----------+----------------+-----------------+--------------------+---------------------+
>
> * It is quite clear that models trained using focal loss outperform those trained using cross entropy loss with smoothed labels both before and after temperature scaling.
>
> * All the networks trained on smoothed labels are able to achieve test set accuracies which are in the state-of-the-art ballpark. Moreover, we observe a significant improvement in both ECE and Ada-ECE values before temperature scaling for models trained using cross entropy loss with smoothed labels as compared to models trained using cross entropy loss with one-hot labels. These improvements however, are not reflected in the ECE and Ada-ECE numbers obtained after temperature scaling.
>
> * It is quite interesting to note that training on smoothed labels causes the models to become less confident on their predictions in general as we often obtain optimal temperatures which are lower than 1. This means that temperature scaling for these models is increasing their confidence. On the other hand, optimal temperatures for models trained using cross entropy with one-hot labels are much greater than 1 and hence, temperature scaling is lowering the confidence of these models.
>
> We will add the numbers obtained from models trained using cross entropy with smoothed labels (both with smoothing factors 0.1 and 0.05) to the paper. We present the preliminary set of results here (and not in the paper) because we need to train other networks (Wide ResNet, DenseNet, etc.) as well to have a complete set of results which can then be included in the paper.

---

> > ### Author Response · Authors · 2019-11-14
> > **Response to R3 Comment 1: Comparison with cross entropy applied to smoothed labels (Part 2)**
> >
> > Table 2
> > +--------+-----------+-----------+------------+----------+----------------+-----------------+--------------------+---------------------+
> > |  Loss  | Smoothing |  Dataset  |   Model    | Error(%) | ECE(%) (Pre T) | ECE(%) (Post T) | Ada-ECE(%) (Pre T) | Ada-ECE(%) (Post T) |
> > +--------+-----------+-----------+------------+----------+----------------+-----------------+--------------------+---------------------+
> > | CE    |       0.0 | CIFAR-10  | ResNet-50  |     4.95  |           4.35  | 1.35(2.5)       |               4.33  | 2.14(2.5)           |
> > | CE    |       0.0 | CIFAR-10  | ResNet-110      4.89   |           4.41  | 1.09(2.8)       |               4.40  | 1.84(2.7)           |
> > | CE    |       0.0 | CIFAR-100| ResNet-50  |    23.30 |          17.52 | 3.42(2.1)       |              17.52 | 3.67(2.1)           |
> > | CE    |       0.0 | CIFAR-100| ResNet-110|    22.73 |          19.05 | 4.43(2.3)       |              19.05 | 5.50(2.4)           |
> > | Focal |     0.0 | CIFAR-10  | ResNet-50  |     4.98   |           1.55 | 0.95(1.1)        |               1.56  | 1.26(1.1)           |
> > | Focal |     0.0 | CIFAR-10  | ResNet-110|     5.42   |           1.87 | 1.07(1.1)        |               2.07  | 1.67(1.1)           |
> > | Focal |     0.0 | CIFAR-100| ResNet-50  |    23.22  |           4.50 | 2.00(1.1)        |               4.39  | 2.33(1.1)           |
> > | Focal |     0.0 | CIFAR-100| ResNet-110|    22.51  |           8.56 | 4.12(1.2)        |               8.55  | 3.96(1.2)           |
> > +-------+-----------+-----------+------------+----------+----------------+-----------------+--------------------+---------------------+

---

> ### Author Response · Authors · 2019-11-14
> **Response to R3 Comment 2: Data Augmentation and Calibration**
>
> We sincerely thank the reviewer for this comment which we address as follows.
>
> A calibrated model without good test set accuracy isn’t useful. Hence, in all our experiments, we set the training parameters in a way so that we get models that generalise well, obtaining state-of-the-art test set accuracies for each model on each dataset. In order to achieve this, we had to apply data augmentation on both the CIFAR-10/100 training sets for all the loss functions, as without that we don’t get state-of-the-art test set accuracies on CIFAR-10/100. The data augmentation involved standard techniques like random crops and random horizontal flips. However, we didn’t have to use any data augmentation for the NLP datasets, 20 Newsgroups and SST Binary to achieve state-of-the-art results on them. This shows that our adaptation of focal loss performed well irrespective of data augmentation.
>
> In order to make these points clear, we have added them in Appendix D, where we describe the training and implementation details for our experiments. Having said that, numerically investigating the importance of data augmentation in the context of model calibration is, in itself, an interesting idea, which we think would make for interesting future work.

---

### Decision · Program_Chairs · 2019-12-19

**Decision:**

Reject

**Comment:**

The paper investigates the effect of focal loss on calibration of neural nets.

On one hand, the reviewers agree that this paper is well-written and the empirical results are interesting. On the other hand, the reviewers felt that there could be better evaluation of the effect of calibration on downstream tasks, and better justification for the choice of optimal gamma (e.g. on a simpler problem setup).

I encourage the others to revise the draft and resubmit to a different venue.